# A multi-proxy analysis of Late Quaternary ocean and climate variability for the Maldives, Inner Sea

Dorothea Bunzel[1], Gerhard Schmiedl[1], Sebastian Lindhorst[1], Andreas Mackensen[2], Jesús Reolid[1], Sarah Romahn[1], Christian Betzler[1]

[1]Center for Earth System Research and Sustainability (CEN), Institute for Geology, University of Hamburg, Hamburg, D-20146, Germany

[2]Alfred Wegener Institute (AWI), Helmholtz Centre for Polar and Marine Research, Bremerhaven, D-27568, Germany

*Correspondence to*: Dorothea Bunzel (dorothea.bunzel@uni-hamburg.de)

**Abstract.** As a natural sediment trap, marine sediments of the sheltered central part of the Maldives Inner Sea represent an exceptional archive for paleoenvironmental and climate changes of the equatorial Indian Ocean. To evaluate the complex interplay between high-latitude and monsoonal climate variability, related dust fluxes, and regional oceanographic responses, we focused on Fe/Al, Ti/Al and Si/Ca ratios as proxies for terrigenous sediment delivery, and total organic carbon (TOC) and Br XRF counts as proxies for marine productivity. Benthic foraminiferal fauna distributions, grain size, and stable $\delta^{18}$O and $\delta^{13}$C data were used for evaluating changes in the benthic ecosystem, as well as changes in the intermediate water circulation, bottom water current velocity and oxygenation.

Our multi-proxy data record reveals an enhanced dust supply during the glacial intervals, causing elevated Fe/Al and Si/Ca ratios, an overall coarsening of the sediment and an increasing amount of agglutinated benthic foraminifera. The enhanced dust fluxes can be attributed to higher dust availability in the Asian desert and loess areas and its transport by intensified winter monsoon winds during glacial conditions. These combined effects of wind-induced mixing of surface waters and dust fertilisation during the cold phases resulted in an increased surface-water productivity and related organic carbon fluxes. Thus, the development of highly diverse benthic foraminiferal faunas with certain detritus and suspension feeders were fostered. The difference in the $\delta^{13}$C signal between epifaunal and deep infaunal benthic foraminifera reveals intermediate water oxygen concentrations between approximately 40 and 100 µmol kg$^{-1}$ during this time. The precessional fluctuation pattern of oxygen changes resembles that from the deep Arabian Sea, suggesting an expansion of the Oxygen Minimum Zone (OMZ) from the Arabian Sea into the tropical Indian Ocean with a probable regional signal of strengthened winter-monsoon-induced organic matter fluxes and oxygen consumption, and further controlled by the varying inflow intensity of the Antarctic Intermediate Water (AAIW). In addition, the bottom water oxygenation pattern of the Maldives Inner Sea reveals a long phase of reduced ventilation during the last glacial period. This process is likely linked to the combined effects of generally enhanced oxygen consumption rates during high-productivity phases, reduced AAIW production, and the restriction of upper bathyal environments of the Inner Sea during sea-level lowstands. Thus, our multi-proxy record reflects a close linkage between the Indian monsoon oscillation, intermediate water circulation, productivity and sea-level changes on orbital time-scale.

## 1 Introduction

Sedimentation and biogeochemical processes of the tropical and subtropical northern Indian Ocean are closely linked to the intensity and seasonal changes of the Indian monsoon system. From June to October, the region is dominated by the southwestern (SW) monsoon, while the northwestern (NE) monsoon operates from December to April (Wyrtki, 1973; Schott and McCreary, 2001). During the SW monsoon, coastal and open-ocean upwelling results in maximum surface water productivity and related organic matter fluxes in the Arabian Sea (Nair et al., 1989, Rixen et al., 1996). At the same time, the Southwest Monsoon Current (SMC) transports high-saline surface waters from the Arabian Sea into the equatorial region (Schott and McCreary, 2001). The SW summer monsoon variability is strongly coherent over the precessional band and

reveals a close but lagged response to northern hemisphere summer insolation maxima (Clemens and Prell, 2003). In comparison, during the NE winter monsoon the ocean current system reverses and the Northeast Monsoon Current (NMC) transports lower-saline surface waters from the Bay of Bengal into the western Indian Ocean. Thus, during the NE monsoon, elevated chlorophyll *a* concentrations are mainly restricted to the Indian west coast and the Maldives area, while chlorophyll *a* strongly decrease in the Arabian Sea at the same time (Sasamal, 2007; de Vos et al., 2014; NASA MODIS-Aqua, 2014).

Evaluations of deep-sea sediment archives from the Arabian Sea delivered not only comprehensive information on the pacing and intensity of the Indian summer monsoon on orbital time scales (Clemens and Prell, 1990; Clemens et al., 1991; Clemens et al., 1996; Clemens and Prell, 2003; Ziegler et al., 2010; Caley et al., 2011a), but also show variance within orbital bands to more rapid climate shifts on suborbital time scale (Schulz et al., 1998; Altabet et al., 2002; Gupta et al., 2003; Pichevin et al., 2007; Böning and Bard, 2009; Caley et al., 2013; Deplazes et al., 2013). Records from the equatorial Indian Ocean provide a more diverse and partly contradictory picture since this region is not only influenced by the summer and winter monsoons but also by the strength of Indian Ocean Equatorial Westerlies (IEW), which are stronger during the intermonsoon seasons in spring and fall and are inversely related to the Indian Ocean Dipole (Hastenrath et al., 1993; Beaufort et al., 2001). Variations in surface water properties at a site close to the Maldives platform revealed maximum productivity at times of enhanced winter monsoon winds associated with precessional maxima in ice volume (Rostek et al., 1997). By contrast, an upper-bathyal benthic foraminiferal record from the Maldives Ridge suggests that Late Quaternary changes in organic matter fluxes are either driven by summer monsoon winds (Sarkar and Gupta, 2009) or linked to changes in the IEW strength (Sarkar and Gupta, 2014). The precessional variability in productivity records from the equatorial Indo-Pacific Ocean has been attributed to the influence of low-latitude insolation on the IEW strength and on long-term dynamics of the El Niño-Southern Oscillation (ENSO) (Beaufort et al., 1997, 2001).

The seasonal reversing circulation pattern of the modern monsoon system established ~12.9 Ma ago, as indicated by the onset of drift deposition in the Maldives Inner Sea (Betzler et al., 2016). Whereas, early aeolian dust deposits at the Maldives between ~20-12.9 Ma can be attributed to a weak proto-monsoon, when prevailing wind systems were strong enough to transport dust material into the northern Indian Ocean, but not sufficient strong to promote the water currents that generate drift deposits (Aston et al., 1973; Guo et al., 2002; Grand et al., 2015; Betzler et al., 2016). Since the early Miocene, the deposition of aeolian particles to the Maldives increased stepwise, coinciding with the onset of Asian desertification derived from loess deposits in northern China (Guo et al., 2002; Betzler et al., 2016). On a regional scale, these changes in aridity were caused by various factors, comprising the Tibetan-Himalayan plateau uplift, which intensified 25-20 Ma ago, and changed wind circulation patterns due to the uplifted topography (Ruddiman and Kutzbach, 1989; Manabe and Broccoli, 1990; Harrison et al., 1992; Guo et al., 2002). However, little detailed information is available about the variability of aeolian dust delivery, its provenance, and linkage to prevailing monsoon circulation patterns in the Maldives area since the Late Quaternary.

In addition to the aeolian influx, sea-level fluctuations and bottom currents played a significant role in the evolution of the sedimentary system of the Maldives (Betzler et al., 2009, 2013a, 2016; Lüdmann et al., 2013). Relative sea-level highstands caused a flooding of the carbonate platforms and an export of shallow-water carbonate sediments into the adjacent basins, whereas sea-level lowstands caused an exposure of the platforms and likely a strengthened restriction of the Inner Sea area (Schlager et al., 1994; Paul et al., 2012; Lüdmann et al., 2013; Betzler et al., 2013b, 2016). Moreover, previous studies have shown that during times of restricted water circulation in the Inner Sea the deposition of organic matter was likewise facilitated (Betzler et al., 2016). In recent times, the decay of organic matter in the water column and the generally reduced ventilation of regional subsurface waters in the Indian Ocean results in the development of a strong Oxygen Minimum Zone (OMZ) between 200 and 1200 m water depth (Reid, 2003; Stramma et al., 2008). This oxygen depletion is fostered by the semi-enclosed nature of the northern Indian Ocean, the long pathway of intermediate water from its main formation sites at 40° S in the central South Indian Ocean (Antarctic Intermediate Water, AAIW) and Indonesia (Indonesian Intermediate Water, IIW) (Olson et al., 1993; You, 1998), and the contribution of low-oxygen outflow waters

from the Red Sea and the Persian Gulf (Jung et al., 2001; Prasad and Ikeda, 2001). The OMZ extends from the Arabian Sea into the equatorial Indian Ocean. While oxygen concentrations in the Arabian Sea can be as low as 0.1 ml l$^{-1}$ (or 5 µmol kg$^{-1}$) (Reid, 2003), they still reach low oxic values of around 1 ml l$^{-1}$ (or 45 µmol kg$^{-1}$) in 500 to 1000 m water depths of the Maldives region (Weiss et al., 1983; Reid, 2003). Phases of an intensified OMZ in the northern Indian Ocean are controlled by both changes in ventilation of intermediate-/deep-water masses (Schmiedl and Leuschner, 2005; Ziegler et al., 2010; Das et al., 2017), and by regional oxygen consumption responding to monsoon-driven upwelling and high surface productivity (e.g. Reichart et al., 1998; Den Dulk et al., 2000; Das et al., 2017). This addresses the ongoing debates, if fluctuations of the OMZ in the Maldives area can be linked to a basin-wide change in the composition of intermediate waters and/or super-regional oceanic processes, respectively. And what is the role of regional sea-level changes in this context.

Here we present a multi-proxy data set on the links between climate variability, ocean circulation, sedimentation and biogeochemical processes of the Maldives Inner Sea. Specifically, our study addresses the following questions: (1) Which impact did orbital-scale changes of the Indian monsoon have on dust fluxes and marine environments of the Maldives? (2) How did global sea-level changes influence the sedimentation processes and benthic ecosystems of the Maldives Inner Sea? (3) Can we trace the influence of changes in the configuration of intermediate waters and how are these changes related to super-regional oceanographic processes? The Maldives Inner Sea is ideally situated to answer these questions because it lies in the central part of the Indian Ocean and is therefore situated in a region influenced the different processes introduced above. The Maldives also appear as an ideal place to trace back paleoceanographic variations in time, as seismic surveys have shown that the Maldives are comparable to a large natural sediment trap with a continuous succession since the Neogene (Betzler et al. 2009, 2013a, 2013b, 2016; Lüdmann et al., 2013).

## 2 Material and Methods

### 2.1 Sediment cores and material descriptions

For this study, two sediment cores were retrieved from different sites in the Inner Sea of the Maldives (Fig. 1). The 5.97 m long sediment core SO236-052-4 was obtained by means of a gravity corer in the framework of R/V SONNE cruise SO236 in August 2014, E of the North Ari Atoll in the central part of the Inner Sea (03°55.09'N; 73°08.48'E) and from a water depth of 382 m. The 12.94 m long sediment core M74/4-1143 is a piston core from R/V METEOR cruise M74/4, obtained in 2007 from E of the Goidhoo Atoll (04°49.50'N; 73°05.04'E) at a water depth of 387 m (Fig. 1). For comparison with the gravity core, this study is focused on the first 6.00 m of core M74/4-1143.

The sediment of core SO236-052-4 consists of an alternation of non-lithified fine-grained ooze with abundant pteropods, sponge spicules, planktonic and benthic foraminifera, and echinoid remains, with minor otolites and fragments of gastropods and bivalves. There are locally some intervals that present up to 4 cm bioclast including solitary corals and thin-shelled bivalves. The entire succession is intensely reworked by bioturbation. Just a few primary structures, usually main boundaries between facies are preserved (i.e. the sharp contacts at 2.35 meters below sea-floor, mbsf, and at 3.90 mbsf). Discrete burrows are scarce. The succession recovered at site M74/4-1143 has been described by Betzler et al. (2013b). The core consists mainly of periplatform ooze containing planktonic foraminifera, pteropods, otoliths, mollusc remains, benthic foraminifers, sponge spicules, and echinoid debris. Down the core, light and dark colored greenish to olive gray intervals alternate (Betzler et al., 2013b).

### 2.2 Geochemical analyses

Scanning X-Ray Fluorescence (XRF) element analysis of core SO236-052-4 was carried out at the MARUM, University of Bremen, using an Avaatech XRF Core Scanner II. Element analysis was performed at 1 cm intervals, using generator settings of 50 kV (1.0 mA current), 30 kV (1.0 mA) and 10 kV (0.2 mA), and a sampling time of 20 seconds per measurement. Raw data spectra were processed using the software package WIN AXIL. Element ratios (Fe/Al, Ti/Al, Si/Ca,

Sr/Ca) were calculated and used for environmental interpretations following Martin et al. (1991), Zhang et al. (1993), Boyd et al. (2000), Gao et al. (2001), Lourens et al. (2001), Jickells et al. (2005), Itambi et al. (2009) and Croudace and Rothwell (2015). Bromine (Br) XRF counts were used as indicator of variability in the marine organic carbon (MOC) content (Ziegler et al., 2008; Caley et al., 2013).

The total inorganic carbon (TIC, calcium carbonate) and total organic carbon (TOC) contents of core SO236-052-4 were measured at 5 cm spacing. The carbon content of the grain-size fraction < 63 μm was determined using a LECO DR144 carbon analyser. All samples were freeze-dried. Subsequently, one subsample was measured at 1350 °C to obtain the total carbon (TC) content. A second subsample was heated to 550 °C for 5 hours to remove the organic carbon prior to measurement in the LECO; this gave the TIC content. The difference between TC and TIC contents is regarded as the TOC content. Calcium carbonate contents were then calculated from the total inorganic carbon content as follows: $CaCO_3$ (%) = (TC - TOC) $\times$ 8.33, where 8.33 is the atomic proportion of carbon in $CaCO_3$ (e.g. Müller et al., 1994; di Primio and Leythaeuser, 1995; Romero et al., 2006).

### 2.3 Grain-size analyses

For bulk grain size analysis, core SO236-052-4 was sampled equidistantly (1.5 cm$^3$ each 1 cm). Samples were wet-sieved (2000 μm) to remove very coarse particles, and subsequently suspended in water with addition of a 0.05 % solution of Tetrasodium Diphosphate Decahydrate as dispersant.

The mean grain size of the non-carbonate fraction between 10 to 63 μm, the sortable silt, has been shown to be a reliable proxy for palaeocurrent strength in predominantly siliciclastic sediments (Manighetti and McCave, 1995; McCave et al., 1995a, 1995b; Hall et al., 1998; Bianchi et al., 1999; McCave and Hall, 2006). The method makes use of the non-carbonate fraction only and is therefore expected to be unaffected by primary carbonate production and burial diagenesis. Samples for the determination of the sortable silt component (c. 20 cm$^3$ each) were taken equidistantly (in core M74/4-1143 at 5 cm intervals down to 1 m core depth and at 10 cm intervals underneath, and at 5 cm downcore intervals in core SO236-052-4). Subsequently to wet-sieving, the fraction < 63μm was cooked in $H_2O_2$ to remove the organic portion, and treated with 1M $Ca_3COOH$ to dissolve the carbonate. Biogene opal was removed with 2M $NaHCO_3$. The remainder was dispersed in water for grain-size determination.

Bulk grain size and sortable silt measurements were done using a Helos KF Magic Laser particle size analyzer and measuring ranges of either 0.5/18-3500 μm (for bulk grain size) or 0.25-87.5 μm (for the non-carbonate fraction). To ensure accuracy of measurements and absence of a long-term instrumental drift, an in-house grain-size standard was measured regularly. Grain size statistics are based on the graphical method (Folk and Ward, 1957) and were calculated using the software GRADISTAT (Blott and Pye, 2001).

### 2.4 Foraminiferal faunal and stable isotope analyses

For stable isotope analyses core SO236-052-4 was sampled at 5 cm spacing, and for benthic foraminiferal faunal analysis at 10 cm spacing. All samples were wet-sieved over a 63 μm screen and the residues subsequently dried at 38 °C. The benthic foraminiferal analysis was carried out on the > 125 μm size fraction and based on allocate splits in order to obtain approximately 300 tests. Genus and species identifications mainly based on Loeblich and Tappan (1988), Hottinger et al. (1993), Jones (1994), Debenay (2012), Milker and Schmiedl (2012) and Holbourn et al. (2013). The genera *Cymbaloporetta* and *Tretomphaloides* were summarized as meroplanktonic benthic foraminifera (BF) since they are known to have planktonic drift phases as part of their dispersal strategy (Banner et al., 1985; Alve, 1999). For analysis based on the test material all individuals of the foraminiferal orders Astrorhizida, Lituolida and Textulariida were summarized as agglutinated BF.

Benthic foraminiferal assemblages were defined by Q-mode Principal Component Analysis (PCA) with varimax rotation using the software SYSTAT, version 5.2.1. Following Schmiedl et al. (1997), only foraminiferal taxa with percentages ≥ 1 %

in at least one sample and/or taxa, which occur at least in two samples were used for the statistical analysis. Loadings $\geq 0.5$ were defined as significant (Backhaus et al., 2008). The Shannon-Wiener diversity index H(S) was calculated after Murray (2006) based on the function H(S) = (-1) $\sum_{i=1}^{s} p_i \times \ln(p_i)$, where $S$ is the species number and $p_i$ the relative abundance of the $i$-th species.

Stable oxygen and carbon isotope records were generated for planktonic and benthic foraminifera. Approximately 10 tests of the planktonic foraminifer *Globigerinoides ruber* (white) were selected from the 250-350 µm size fraction of core SO236-052-4. Stable isotope data of *G. ruber* (white) of core M74/4-1143 were taken from Betzler et al. (2013b). In addition, approximately 2-5 tests of the epibenthic foraminifera *Cibicides mabahethi* and of the deep infaunal species *Globobulimina affinis* s.l. were selected from the size fraction > 125 µm of core SO236-052-4. Stable oxygen and carbon isotope analyses were performed with Finnigan MAT253 gas mass spectrometer coupled to automatic carbonate preparation device Kiel IV, respectively. The mass spectrometer was calibrated to the PDB scale via international standard NBS19, and results are given in $\delta$-notation versus VPDB. Based on measurements of an internal laboratory standard (Solnhofen limestone) together with samples over a 1-year period, precision was better than 0.08 ‰ for $\delta^{18}$O and 0.06 ‰ for $\delta^{13}$C, respectively.

For core SO236-052-4 changes in bottom water oxygen concentrations were estimated based on the $\delta^{13}$C difference between the epifaunal (*C. mabahethi = Cm*) and deep infaunal (*G. affinis* s.l. = *Ga*) benthic foraminifera using the function $\Delta\delta^{13}$C = 0.00772 × [O$_2$] + 0.41446, wherein [O$_2$] concentrations between 55 and 235 µmol kg$^{-1}$ have shown a strong linear relationship between bottom water [O$_2$] and $\Delta\delta^{13}$C (Hoogakker et al., 2015). We also applied this function to the low [O$_2$] values ([O$_2$] < 55 µmol kg$^{-1}$), because such a relationship may also exist for low [O$_2$] values and $\Delta\delta^{13}$C, although not investigated by Hoogakker et al. (2015). For [O$_2$] reconstruction two $\delta^{13}$C values of core SO236-052-4 (at 0.17 and 130.92 ka) have not been considered due to the weak validity in the $\delta^{13}C_{Ga}$ signal. For comparison, oxygen concentration changes of a deep-sea sediment core from an Arabian Sea site (GeoB3004, 1803 m water depth) were taken from Schmiedl and Mackensen (2006). These data are based on the difference between the epifaunal *Cibicidoides wuellerstorfi* (*Cw*) and *G. affinis* (*Ga*). Further, the $\delta^{13}$C gradient between *G. ruber* (white; *Gr*) and *C. mabahethi* (*Cm*) of core SO236-052-4 was estimated to assess the sea surface and bottom-water stabile carbon isotope difference and water column mixing.

**2.5 Spectral analyses**

For the evaluation of periodic temporal variability Blackman-Tukey spectral analyses were performed for TOC, Fe/Al and Ti/Al ratios, Br XRF counts of core SO236-052-4, and the reconstructed oxygen concentrations of core SO236-052-4 and Arabian Sea core GeoB3004 (Schmiedl and Mackensen, 2006), in comparison to the Δ-insolation at 30° N. The analyses were carried out with the AnalySeries software, version 2.0, 05/2005 (Paillard et al., 1996). Prior to the analyses, the records of SO236-052-4 were rescaled with Δt = 2 ka (TOC, oxygen) and Δt = 0.5 ka (Fe/Al, Ti/Al, Br XRF counts), and the oxygen record of GeoB3004 with Δt = 1 ka.

**2.6 Radiocarbon dating and compilation of the age model**

Accelerator Mass Spectrometry (AMS) radiocarbon dating was carried out at the Beta Analytic Radiocarbon Dating Laboratory on mixed surface-dwelling planktonic foraminifera from 35 cm, 80 cm and 140 cm depth of core SO236-052-4 (Table 1). Due to the contrasting available reservoir age correction values (varying between 301 to 544 years; Dutta et al., 2001; Southon et al., 2002), the AMS $^{14}$C ages were corrected for the global reservoir age of 400 years in order to minimize potential errors and converted to calendar years using the radiocarbon calibration program CALIB (version 7.0.4; Stuiver and Reimer, 1993) and the calibration curve Marine13 (Reimer et al., 2013). Additional age tie points were derived from graphical correlation of the benthic $\delta^{18}$O record of core SO236-052-4 with the LR04 standard benthic stack (Lisiecki and

Raymo, 2005) using the software AnalySeries 2.0 (version 5/2005; Paillard et al., 1996). The age model of core M74/4-1143 (Betzler et al., 2013b) was revised by graphical correlation with the planktonic $\delta^{18}$O record of core SO236-052-4 (Fig. 2).


We will provide all data records of core SO236-052-4, which were used and discussed in this manuscript, to the PANGAEA Open Access library.

## 3 Results

### 3.1 Age model and sedimentation rate

Based on the radiocarbon ages and the alignment of the stable oxygen isotope stratigraphy, core SO236-052-4 comprises sediments of the past 207.7 ka, respectively (Fig. 2). The top 6 m of sediment core M74/4-1143 comprise the past 242.3 ka. Average sedimentation rates varied between 3.5 cm ka$^{-1}$ in SO236-052-4 and 4.4 cm ka$^{-1}$ in M74/4-1143. Maximum sedimentation rates occurred during interglacial intervals, with 6.8 cm ka$^{-1}$ (SO236-052-4) and 8.4 cm ka$^{-1}$ (M74/4-1143) for the Eemian, and 7.1 cm ka$^{-1}$ (SO236-052-4) to 3.0 cm ka$^{-1}$ (M74/4-1143) for the Holocene (Fig. 2).

### 3.2 Foraminiferal stable oxygen and carbon isotope records

The $\delta^{18}$O values of core SO236-052-4 vary between -3.09 and -0.68 ‰ in the planktonic *G. ruber*, between 0.91 and 2.51 ‰ in the epibenthic *C. mabahethi*, and between 1.10 and 5.02 ‰ in the deep infaunal *G. affinis* (Fig. 3). Generally, the $\delta^{18}$O records reveal a consistent picture with relatively higher values during glacial intervals and lower values during interglacial intervals. The $\delta^{13}$C values of core SO236-052-4 vary between 0.12 and 1.25 ‰ in *G. ruber,* between 0.22 and 0.91 ‰ in *C.*
*mabahethi*, and between -0.84 to 0.27 ‰ in *G. affinis* (Fig. 3). Despite considerable short-term variability, all records reveal a stepwise increase of $\delta^{13}$C values with lowest values during the Marine Isotope Stage (MIS) 6 and highest values during the Holocene (Fig. 3).

Considering the past bottom-water oxygen concentration reconstruction for core SO236-052-4 by using the $\Delta\delta^{13}C_{Cm-Ga}$ estimation, the [O$_2$] values varied between 10.43 µmol kg$^{-1}$ ($\triangleq$ 0.50 ‰ $\Delta\delta^{13}C_{Cm-Ga}$) and 139.45 µmol kg$^{-1}$ ($\triangleq$ 1.49 ‰
$\Delta\delta^{13}C_{Cm-Ga}$) with average values of approximately 67.10 µmol kg$^{-1}$ ($\triangleq$ 0.93 ‰ $\Delta\delta^{13}C_{Cm-Ga}$). In addition, a long-lasting oxygen depletion was observed, starting from the end of MIS 5 to the end of MIS 3 (duration of ~50 ka), and with average oxygen concentrations of approximately 52.50 µmol kg$^{-1}$. The oxygen concentration of sediment core GeoB3004 (W Arabian Sea) showed an average oxygen content of approximately 81.75 µmol kg$^{-1}$ ($\triangleq$ 1.05 ‰ $\Delta\delta^{13}C_{Cw-Ga}$), with variations between 22.04 µmol kg$^{-1}$ ($\triangleq$ 0.58 ‰ $\Delta\delta^{13}C_{Cw-Ga}$) and 133.87 µmol kg$^{-1}$ ($\triangleq$ 1.45 ‰ $\Delta\delta^{13}C_{Cw-Ga}$) (Schmiedl and Mackensen, 2006). Spectral
analyses of the oxygen records of cores SO236-052-4 and GeoB3004 reveal strong power in the precession band (Fig. 4a). The comparatively long lasting oxygen depletion during the last glacial period at site SO236-052 is not observed at site GeoB3004.

The $\Delta\delta^{13}C_{Gr-Cm}$ calculation for site SO236-052 showed maximum differences of the planktonic and epibenthic stabile $\delta^{13}$C values (0.68 ‰) during the full interglacial periods of MIS 7, 5 and 1, coinciding with global sea-level highstands.
Accordingly, minimum differences (-0.54 ‰) were documented for the glacial periods MIS 6 and 2 and sea-level lowstands.

### 3.3 Sedimentological and geochemical records

The detailed sedimentological and geochemical data of core SO236-052-4 reveal a glacial-interglacial pattern but also considerable short-term variability (Figs. 4c-f, 5). Sortable silt records are available for both sites and in general they show coarser means during interglacial times, up to 14 µm during the Holocene and > 13 µm during the Eemian (Figs. 5a, 6).
However, absolute values and variability are much greater in core M74/4-1143 which is located in the drift of the Kardiva Channel (see: Betzler et al., 2013b) compared to core SO236-052-4 from the more sheltered part of the Inner Sea. In core

M74/4-1143, sortable silt shows an increase towards the MIS 6/5 transition (Termination II) followed by generally elevated values during the Eemian (maximum 27.17 μm), whereas there is much less variability at the same time in the data from core SO236-052-4. Both cores show a coarsening of the sortable silt towards the MIS 1.

Bulk mean grain size shows a pronounced glacial-interglacial variability with up to 57 μm during the MIS 2 and MIS 6 and values between 10 and 30 μm for the remainder (Fig. 5a). Finest sediments occur during the early MIS 5 and the MIS 1 (10 μm).

Total organic carbon (TOC) and calcium carbonate ($CaCO_3$) contents of core SO236-052-4 reveal reverse glacial-interglacial trends with maximum TOC content during glacial- and maximum carbonate contents during interglacial periods. TOC varies between 0.85 wt. % (interglacial) and 2.06 wt. % (glacial), whereas the carbonate content varies between 77.03 wt. % (glacial) and 89.40 wt. % (interglacial) (Fig. 5b).

Both, the Fe/Al and Ti/Al records, show similar patterns with relatively higher values during glacial periods and with slightly increasing values towards today. The Br XRF counts and the Si/Ca ratio show similar distinct glacial-interglacial patterns, comparable to the TOC record, with generally higher values during glacial periods and lower values during interglacial periods (Fig. 5d). Further, the Si/Ca record is characterized by an abrupt and short-lasting maximum at Termination II (Fig. 5d). Inverse patterns are observed for the Sr/Ca record, which follows the inversed $\delta^{18}O$ curve and shows higher values during interglacial and lower values during glacial periods (Fig. 5d).

Spectral analyses of the TOC, Ti/Al and Br reveal significant but considerably weaker variability in the precession band when compared to the reconstructed oxygen records (Fig. 4c, e). The Fe/Al record lacks substantial precessional variability but is rather dominated by long-term glacial-interglacial changes (Fig. 4e).

### 3.4 Benthic foraminiferal record

In sediment core SO236-052-4, a total of 256 different benthic foraminiferal species were distinguished, with 51 to 93 different species per sample. The diversity H(S) is relatively high and varies between 3.3 and 4.0, with a slight long-term decrease towards today (Fig. 7a). The foraminiferal fauna at the Inner Sea is dominated by hyaline taxa. In comparison, agglutinated individuals were less abundant, but with increasing relative abundances up to approximately ≥ 20 % of the entire fauna during the glacial periods (Fig. 8e). The three-component model of the Q-mode PCA explains 89.14 % of the total variance (Table 2). Assemblage 1 (PC1) explains 31.54 % of the total variance and is dominated by *Neouvigerina proboscidea* and *Discorbinella araucana,* with *Hyalinea inflata*, *Cymbaloporetta squammosa*, *Bulimina marginata* and *Rosalina vilardeboana* as associated taxa (Table 2). This assemblage occurred mainly during the late MIS 7 and 5 and is less pronounced during MIS 4 to early MIS 2 (Fig. 7b). Assemblage 2 (PC2) explains 30.54 % of the total variance and is dominated by *Cibicides mabahethi,* with *Discorbinella bertheloti*, *Siphogenerina columellaris*, *Gyroidina umbonata, Reophax* sp.*, H. inflata*, and *D. araucana* as associated taxa (Table 2). Assemblage 2 occurred mainly during MIS 6 and 1 (Fig. 7c). Assemblage 3 (PC3) explains 27.06 % of the total variance and is dominated by *N. proboscidea* and *Hoeglundina elegans,* with *D. bertheloti*, *Cibicidoides subhaidingeri*, *Discorbis* sp., *Spiroplectinella sagittula* s.l., and *C. mabahethi* as associated taxa (Table 2). Assemblage 3 occurred during MIS 4 to 2 (Fig. 7d). The distribution of the most important benthic foraminiferal species, which characterize the three faunal assemblages, are displayed in Fig. 7e-g. The most abundant species include *C. mabahethi* (maximum relative abundance of ~17 % during MIS 6), the weakly hispid *N. proboscidea* (maximum of ~16 % at the end of MIS 5) and *D. araucana* (maximum of ~11 % during the onset of MIS 5). Meroplanktonic benthic foraminifera (genera *Cymbaloporetta* and *Tretomphaloides*) occurred in elevated numbers during interglacial periods (maximum of 10.88 %), particularly during MIS 5 (Figs. 6d, 7e).

### 4 Discussion

### 4.1 Dust fluxes, marine productivity and monsoon variability

The Ti/Al and Fe/Al in core SO236-052-4 were used as proxies for terrigenous sediment delivery, reflecting changes in the deposition of aeolian dust and thus in the aridity of the hinterland/source area of the dust (Zhang et al., 1993; Lourens et al., 2001; Itambi et al., 2009). Local sources of Fe-rich sediments can be excluded since the sediments of the Maldives archipelago and adjacent shallow- to deep-water environments are characterized by carbonates comprising reef and lagoon carbonates of the islands and pelagic deep-water carbonates (Betzler et al., 2013b; Reolid et al., 2017). This is also reflected by the calcium carbonate content, which is very high at site SO236-052 throughout the past 200 ka (Fig. 5b). Studies on modern aerosols of the North Pacific region indicate that most of the oceanic iron input is derived from atmospheric transport after mobilisation from the central Asian deserts and Chinese loess plateau (Duce and Tindale, 1991). Accordingly, the most likely dust sources for the observed Fe in the sediments of the Maldives are the Indian subcontinent and the Asian desert and loess areas (Roberts et al., 2011), although a minor contribution from northeast Africa and Arabia cannot be excluded (Chauhan and Shukla, 2016). The latter regions rather have been identified as major dust sources in the Arabian Sea based on specific clay mineral composition and magnetic susceptibility of the lithogenic fraction (deMenocal et al., 1991; Sirocko and Lange, 1991). Northwest winds over the Arabian Peninsula blow dust into the Arabian Sea, when the Arabian source areas undergo aridification during high-latitude ice cover (deMenocal et al., 1991; Sirocko and Lange, 1991). In the northwestern Arabian Sea, associated dust fluxes are in phase with maximum ice volume but also vary in the precessional band, suggesting a likewise strengthening of the SW summer monsoon as an important regional driver (Clemens and Prell, 1990; Clemens et al., 1996). In contrast, in the eastern equatorial Indian Ocean the majority of dust is transported via the NE monsoon, coinciding with the prevailing wind system, which blows during northern hemisphere winter. Therefore, elevated Fe/Al and Ti/Al ratios at site SO236-052 indicate the combined effects of enhanced glacial dust availability in the source areas and dust transport to the Maldives with generally strengthened NE monsoon winds during the glacial intervals of MIS 6 and MIS 4-2. On a global scale, the generally colder and drier glacial conditions resulted in a two- to fivefold increase of dust fluxes (Maher et al., 2010).

The observed response of the winter circulation of the Maldives to glacial conditions is in line with the finding of a general strengthening of the NE Indian monsoon after initiation of the northern hemisphere glaciation (Gupta and Thomas, 2003). But our dust proxies show little (Ti/Al) or no (Fe/Al) variability at the precessional band (Fig. 4e), which should be expected if the dust fluxes were directly proportional to the intensity of the winter monsoon (Caley et al., 2011a, b). For a statistically more robust evaluation of the full orbital variability (including the long-wave components) considerably longer time series would be required. Nevertheless, graphical comparison of our data series reveals a dominant eccentricity component in the dust proxies with pronounced glacial-to-interglacial changes, suggesting a link to high-latitude climate and environmental changes (Fig. 8). Therefore the dust records of the Maldives Inner Sea are mainly driven by the generally enhanced dust availability during glacial intervals. As a major dust source, the Chinese loess plateau is strongly influenced by the East Asian Monsoon (EAM). During the Late Quaternary, EAM and related vegetation changes are characterized by predominant eccentricity cycles associated with the advance and retreat of the boreal ice sheets (Ding et al., 1995; Liu et al., 1999; Sun et al., 2006; Hao et al., 2012). Our conclusion is also in line with the lithogenic flux reconstructions from the Arabian Sea (e.g. site ODP722; Clemens et al., 1996), which show striking changes on the glacial-to-interglacial timescale (in the eccentricity band) and suggest a close relation between high-latitude climate and aridity of the dust source areas. On the Chinese loess plateau, the onset of glacial conditions led to an abrupt increase of atmospheric dust loadings (Zhang et al., 2002), suggesting the operation of climate-vegetation feedbacks. Enhanced deposition of terrestrial particles at site SO236-052 led to a generally coarsening of the glacial sediment and fostered the distribution of agglutinated benthic foraminifera, which reached a relative abundance of up to ~20 % during the last glacial period (Figs. 5, 8e). Most agglutinated foraminifera in core SO236-052-4 belong to the Textulariida, such as *Spiroplectammina sagittula*, *Textularia calva* or *Textularia pala*. These and related taxa are often associated with relatively coarse-grained substrates and they preferentially use siliciclastic grains for building up their test walls (Allen et al., 1999; Murray, 2006; Armynot du Châtelet et al., 2013).

The modern equatorial Indian Ocean is limited in the micronutrient iron (Wiggert et al., 2006; Maher et al., 2010) and therefore, enhanced aeolian Fe fluxes to the ocean during glacial periods likely have a direct impact on the seasonal surface productivity (Martin et al., 1991; Boyd et al., 2000; Gao et al., 2001; Jickells et al., 2005), which also influences the deep-sea benthic ecosystems through seasonal phytodetritus pulses. Similar relations between Fe fluxes and surface ocean productivity have been reported from the Southern Ocean (Anderson et al., 2014; Martínez-García et al., 2014) and the equatorial Pacific Ocean (Costa et al., 2016). The TOC and Br contents of marine sediments are widely used as proxies for organic matter fluxes and surface water productivity (Müller and Suess, 1979; Rühlemann et al., 1999; Ziegler et al., 2008 2009; Caley et al., 2013). But the applicability of both proxies in quantitative reconstructions is limited by the specific sedimentological and biogeochemical processes at the sediment-water interface, including the bulk accumulation rate and bottom water oxygenation (Möbius et al., 2011; Schoepfer et al., 2015; Naik et al., 2017). However, elevated TOC and Br values in core SO236-052-4 suggest generally enhanced organic matter fluxes during glacial periods, which may reflect the influence of Fe fertilisation (Fig. 8b-d).

The benthic foraminiferal faunas at site SO236-052 reveal a marked glacial-interglacial pattern (Figs. 7, 8f). The diversity, microhabitat partitioning and species composition of deep-sea benthic foraminiferal faunas is mainly controlled by the combined influences of quantity and quality of food supply and oxygen content of the bottom and pore waters (Jorissen et al., 1995; Fontanier et al., 2002). The diversity of the faunas is high, with H(S) values always > 3.2, throughout the studied time interval, suggesting the absence of extreme environmental conditions at the sea floor of the study site. Therefore, the observed faunal changes likely reflect variations in the amount and quality of food supply. The most abundant species of the three benthic foraminiferal assemblages comprise *C. mabahethi, N. proboscidea* and *D. araucana*, all with PC scores > 3 in at least one assemblage (Table 2). Microhabitat studies demonstrated that most species of the genera *Cibicides* and *Cicididoides* live as suspension feeders on or elevated above the sea floor (Lutze and Thiel, 1989; Linke and Lutze, 1993), therefore we assume a similar microhabitat preference for *C. mabahethi*. In the Red Sea, this species is adapted to relatively high oxygen contents and low organic matter fluxes (Edelman-Furstenberg et al., 2001; Badawi et al., 2005). The cosmopolitan *N. proboscidea* inhabits an epifaunal to very shallow infaunal microhabitat (Fontanier et al., 2002; Licari et al., 2003) and has been described as detritus feeder from various bathyal and abyssal environments. In the South Atlantic Ocean, *N. proboscidea* is associated with well-ventilated and oligotrophic conditions (Schmiedl et al., 1997). Whereas this species thrives under moderate to high organic matter fluxes and oxygen-depleted intermediate waters in the Indian Ocean (Murgese and De Deckker, 2007; De and Gupta, 2010) and was used as a proxy for the strength of the SW monsoon (Gupta and Srinivasan, 1992; Gupta and Thomas, 2003; Sarkar and Gupta, 2014). These observations and the high relative abundance of *N. proboscidea* in core SO236-052-4 during the last glacial intervals MIS 4-2, as well as the interglacial interval MIS 5 suggest an adaptation to a wide range of trophic conditions and confirms its tolerance to moderate oxygen depletion. Little information is available on the ecology of *D. araucana* but its flat trochospiral test morphology and its distribution in the North Atlantic Ocean suggest an epifaunal microhabitat and adaptation to suspended food sources (Corliss and Chen, 1988; Koho et al., 2008). Similar to the closely related *D. bertheloti*, it may prefer oxic conditions (Duleba et al., 1999; Smith and Gallagher, 2003), with a tolerance to moderate oxygen depletion (Edelmann-Furstenberg et al., 2001). The shallow infaunal *Hoeglundina elegans*, which mainly occurs together with *N. proboscidea* in assemblage 3 during MIS 4-2, is commonly associated with low to moderate organic matter fluxes, fresh phytodetritus and high oxygen contents (Corliss, 1985; Koho et al., 2008).

The ecological preferences of the dominant taxa suggest that faunal changes at site SO236-052, although pronounced, were driven by rather subtle changes in the amount of organic matter fluxes. Instead, the faunal changes likely reflect variations in lateral suspension of food particles, substrate-specific development of infaunal niches, and the influence of oxygen changes on the quality of the organic matter. The high dominance of the detritus feeders *N. proboscidea* and *H. elegans* in assemblage 3 reflect highest organic matter fluxes during the last glacial MIS 4-2 (Fig. 8f). In contrast, the dominance of the epifaunal suspension feeder *C. mabahethi* in assemblage 2 during the penultimate glacial (MIS 6) suggests

relatively lower organic matter fluxes. The *N. proboscidea/D. araucana* assemblage 1 of MIS 5 reveals some similarity to assemblage 3 but the high abundance of *D. araucana* suggests an overall lower food flux with a considerable amount of suspended particles. In addition, the relatively finer-grained substrate likely opened infaunal niches as indicated by the presence of the shallow to deep infaunal *Bulimina marginata* during the MIS 5 (Jorissen and Wittling, 1999) (Fig. 7e). Contrasts in the boundary conditions between different glacials and interglacials may account for the inconsistent association of certain assemblages to either glacial or interglacial periods. For instance, glacial boundary conditions during MIS 6 were different from MIS 4-2, in which the latter was characterized by relatively higher global ice volume and related lower sea-level (Rohling et al., 2009) and pronounced millennial-scale variability (Dansgaard et al., 1993). Previous studies showed similar patterns, with certain benthic foraminiferal assemblages occurring both during glacial and interglacial periods, e.g. in the Red Sea (Badawi et al., 2005). At the Maldives Inner Sea glacial-to-interglacial changes in food fluxes were likely not extreme and therefore ecological thresholds for certain species and faunas may not have always been passed during glacial-interglacial transitions. A detailed inspection of assemblage 2 (*C. mabahethi*-fauna) actually displays faunal differences between their occurrences in MIS 1 and MIS 6 although *C. mabahethi* is the dominant taxon in both intervals.

While the benthic foraminiferal fauna basically varies on the glacial-interglacial time scale, the TOC, Br and Ti/Al records reveal additional variability in the precessional band (Fig. 4). This suggests that marine environmental conditions in the Maldives are linked to high-latitude climate variability but also to regional monsoonal changes. The surface water productivity of the northern Indian Ocean is driven by wind-induced mixing of the upper water column and upwelling of nutrient-rich subsurface waters and thus reveals a close association with seasonal changes of the monsoonal wind system (Nair et al., 1989). Accordingly, productivity changes in the northern and northwestern Arabian Sea are coherent to the strength of the SW monsoon (Ivanova et al., 2003; Leuschner and Sirocko, 2003; Singh et al., 2011), and along the Indian west coast to the strength of the NE monsoon (Rostek et al., 1997; Singh et al., 2011). The elevated TOC and Br values at site SO236-052 during phases of reduced northern hemisphere summer insolation suggest a direct influence of the Indian winter monsoon on productivity and related organic matter fluxes of the Maldives Inner Sea during the past 200 ka (Fig. 8), which is consistent with the present-day situation (de Vos et al., 2014). The link between the winter monsoon intensity and surface water productivity in the study area is confirmed by the difference between the $\delta^{13}$C values of the epipelagic *G. ruber* and the epibenthic *C. mabahethi* (Fig. 9). Low $\Delta\delta^{13}C_{Gr-Cm}$ values indicate enhanced vertical mixing of the water column, which is associated with increased supply of nutrients from subsurface waters into the photic zone, based on enhanced surface water productivity.

### 4.2 Sea-level changes, sedimentation processes and benthic ecosystem dynamics

The close association of changes in sediment composition (i.e. bulk grain size, carbonate content) at site SO236-052 with the LR04 stable benthic isotope stack (Lisiecki and Raymo, 2005) suggests a dominant influence of sea-level changes on the depositional environments of the Maldives Inner Sea. This is also corroborated by the Sr/Ca variations in the core. In periplatform ooze, i.e. areas around shallow water carbonate banks, higher Sr contents are a consequence of higher input of shallow water aragonite (Dunbar and Dickens, 2003), which is produced in the neritic parts of the platforms and exported to the areas around the platform by currents.

As shown by previous studies, variations in the total organic carbon content in the Inner Sea sediments are considerably triggered by sea-level and ocean current changes (Betzler et al., 2016). Thus, the observed changes in bottom currents likely influenced the lateral transport of suspended organic particles as it is suggested by variations in the relative abundance of suspension feeders in the different benthic foraminiferal assemblages (Figs. 7, 8, Table 2). The dominance of *D. araucana* during MIS 5 and *C. mabahethi* during MIS 6 and MIS 1 indicates phases of enhanced lateral food supply, which for the interglacial periods (MIS 5, MIS 1) correlate with reconstructed higher current velocities and sea-level highstands (Fig. 6). This is shown by the higher sortable silt data at site SO236-052, implying higher bottom current velocities, and which is also supported by the higher amplitude of change and the sortable silt values of the drift deposits recovered by core M74/4-1143.

The different sortable silt amplitudes of both settings in the Maldives are due to the restriction of the central part of the Inner Sea (core SO236-052-4), whereas in comparison the deposition area in the Kardiva Channel (core M74/4-1143) is known to be exposed by much stronger current regimes since the Late Pleistocene (Betzler et al., 2013b; Reolid et al., 2017).

The interglacial intervals (mainly MIS 5 and MIS 7, Fig. 6) contain high abundances of meroplanktonic benthic foraminifera (*Cymbaloporetta*, *Tretomphaloides*), which build floating chambers for dispersal (Banner et al., 1985; Alve, 1999). These taxa are commonly found in shelf environments (Milker and Schmiedl, 2012). Their acme during the last interglacial maximum at upper bathyal depth of the Maldives Inner Sea coincides with an almost absence of other displaced species from reef and lagoon environments, such as *Elphidium*, *Amphistegina* or *Operculina* (Parker and Gischler, 2011). This implies a repeated colonization of upper bathyal environments with meroplanktonic taxa from submerged neritic environments during sea-level highstands and conditions of the strengthened bottom water velocity.

### 4.3 Changes in intermediate water circulation and oxygenation

The epibenthic stable carbon isotope record of core SO236-052-4 lacks a coherent glacial-interglacial pattern but reveals an overall $\delta^{13}C_{Cm}$ increase of ~0.5 ‰ over the past 200 ka (Fig. 10). Long-term trends of similar magnitude have been recorded from sites in the southwestern Pacific Ocean, which were particularly bathed by the well-oxygenated Antarctic Intermediate Water mass (AAIW) during warm intervalls (Thiede et al., 1999; Pahnke and Zahn, 2005; Elmore et al., 2015; Ronge et al., 2015) (Fig. 10). The general resemblance of relative changes in epibenthic $\delta^{13}C$ records from different regions suggests a significant and super-regional role of the AAIW in ventilation of upper bathyal environments of the Maldives Inner Sea, which is consistent with the modern oceanographic situation (You, 1998).

Following the approach of Hoogakker et al. (2015) we estimated the changes in the oxygen content of the intermediate water mass of the Maldives Inner Sea based on the $\Delta\delta^{13}C_{Cm-Ga}$ signal, i.e. the difference between the $\delta^{13}C$ values of the epifaunal *C. mabahethi* and the deep infaunal *G. affinis* s.l. The resulting $[O_2]$ concentrations display significant power in the precession band (23 ka period), with oxic and low oxic conditions related to northern hemisphere insolation maxima and minima, but they never dropped substantially below 45 µmol kg$^{-1}$ ($\approx$1 ml l$^{-1}$) (Fig. 9). Moreover, the oxic to low oxic conditions did not seem to pose stress to the benthic foraminiferal fauna. Instead, the proportion of the deep infauna increases exponentially under dysoxic conditions, i.e. at $[O_2]$ values significantly below 1 ml l$^{-1}$ (Jorissen et al., 2007). The lack of dysoxic conditions at site SO236-052 at any time of the past 200 ka is corroborated by the persistent high diversity across glacial and interglacial periods and the low abundance of deep infaunal taxa. However, the reconstructed $[O_2]$ changes in intermediate waters at site SO236-052 resemble those from the deep OMZ of the western Arabian Sea, which is influenced by the advection of oxygen-rich North Atlantic Deep Water (NADW) (Schmiedl and Mackensen, 2006). The dependence of oxygen changes in Indian Ocean water masses from the inflow of Atlantic and Antarctic water masses is corroborated by a number of recent observations from the northwestern and southeastern Arabian Sea (Pattan and Pearce, 2009; Das et al., 2017; Naik et al., 2017). We therefore assume that the OMZ variability in the Maldives Inner Sea is influenced by the overall strength and lateral southward expansion of the Arabian Sea OMZ, by local monsoon-related organic matter fluxes and oxygen consumption, but it is additionally controlled by the ventilation of southern-derived oxygen-rich intermediate waters (AAIW). The ocean-wide linkage of intermediate water ventilation can be assumed, due to the general resemblance of our epibenthic stable carbon isotope record with comparable records from other areas. On the other hand, the significant variability of our new oxygen reconstruction from the Maldives Inner Sea in the precession band and its resemblance with the reconstruction from the Arabian Sea suggests an additional influence of monsoon-driven biogeochemical processes. The resulting changes in the biogeochemical processes at site SO236-052 are illustrated by the establishment and long-term persistence of the benthic foraminiferal assemblage 3 underlining the positive response of *N. proboscidea* and associated species such as *H. elegans* and *D. bertheloti* to moderately reduced oxygen and increased food levels.

The long period of lowered [$O_2$] values below 60 µmol kg$^{-1}$ centred at MIS 4-3 coincides with a marked monsoon and upwelling maximum in the Arabian Sea (Hermelin and Shimmield, 1995; Clemens and Prell, 2003; Leuschner and Sirocko, 2003; Caley et al., 2011a, b), which caused a strengthening and deepening of the OMZ (Almogi-Labin et al., 2000; Den Dulk et al., 2000; Schmiedl and Leuschner, 2005). The expansion of the Arabian Sea OMZ southward into the equatorial region likely preconditioned the oxygen levels of intermediate waters of the Maldives Inner Sea. There, oxygen minima were

further lowered by the reduced glacial advection of the oxygen-rich AAIW and enhanced regional microbial oxygen consumption, reflecting a superposition of high and low-latitude climate signals. Additionally, an abrupt [$O_2$] drop occur at the end of the last glaciation suggesting a short phase of reduced AAIW advection or increased surface water productivity and related oxygen consumption at depth (Fig. 9).

        The present OMZ of the northwestern Indian Ocean extends from the northern Arabian Sea into the tropical Indian

Ocean (Reid, 2003), reflecting the reduced ventilation of intermediate water masses (due to its remote position) and the biogeochemical processes related to monsoon-induced organic matter fluxes and microbial oxygen consumption. Indeed, the oxygen concentrations in the Indian Ocean display a gradient with very low values in the northern Arabian Sea and increasing values to the South. This gradient illustrates a clear relation to the monsoon-related biogeochemical processes in the Arabian Sea, but is also a reflection of the remote position of the Arabian Sea in terms of intermediate water ventilation.

Nevertheless, a monsoon-induced strengthening of the OMZ in the Arabian Sea (as during MIS 3) will results in an increase of the north-south oxygen gradient in the entire northwestern Indian Ocean, which should then also be detected in the Maldives Inner Sea at a lower amplitude.

        Evaluations of calcareous nannoplankton records, used as indicators for surface productivity, from sediment cores of the equatorial Indian and Pacific Ocean reveal significant variability in the precession band and are coherent and in phase with

February equatorial insolation (Beaufort et al., 1997, 2001). This orbital pattern suggested a close link of equatorial Indian Ocean productivity to the strength of the Indian Ocean Equatorial Westerlies (IEW) and an ENSO-like forcing of equatorial surface ocean productivity during the Late Quaternary (Beaufort et al., 2001). Accordingly, regional productivity and organic matter fluxes in the wider Maldives area may have also been influenced by changes in the strength of the IEW. However, a strong IEW influence in the Maldives Inner Sea is questioned by the low precessional variability observed in our

productivity proxies and the close association of modern phytoplankton blooms of this region with the NE monsoon (Sasamal, 2007; de Vos et al., 2014). To summarize, our new results imply that on orbital time scales changes of the winter monsoon and AAIW advection seem to exert the dominant influence on upper bathyal environments of the Maldives Inner Sea.

**5 Conclusions**

The integrated evaluation of sedimentological, geochemical and micropaleontological proxy records from the Maldives Inner Sea (tropical Indian Ocean) furthers our understanding of links between equatorial climate variability, sea-level changes, changes in intermediate water ventilation and benthic ecosystem dynamics on orbital time scales during the past 200 ka. The main conclusions are:

        (1) Aeolian dust fluxes were considerably enhanced during glacial intervals (MIS 6 and MIS 4-2) as indicated by

increased Fe/Al, Ti/Al and Si/Ca ratios, generally coarsening of the bulk sediment, and increased abundance of agglutinated benthic foraminiferal taxa, which use the siliciclastic grains for test formation. The enhanced dust input was linked to phases of generally increased atmospheric dust loads and NE winds, suggesting a close linkage of Maldives marine environments to the aridity of the central Asian loess areas and the strength of the Indian winter monsoon.

        (2) Increased vertical mixing during glacial phases of intensified winter monsoon resulted in enhanced surface water

productivity and associated organic carbon fluxes to the sea-floor as indicated by TOC values and composition of the benthic foraminiferal fauna. The *Cibicidoides mabahethi* (assemblage 2) and *Neouvigerina proboscidea* (assemblage 3) faunas

dominate during MIS 6 and MIS 4-2 respectively, suggesting differences in the amount and quality of the food delivery for the two glacial intervals. The Br XRF counts and the TOC record reveal additional variability on the precessional band (as shown with the Blackman-Tukey spectral analysis, Fig. 4), which are inversely correlated to northern hemisphere summer insolation underlining a close link of regional vertical mixing of the water column and marine productivity to the Indian winter monsoon.

(3) Glacial-interglacial changes in sea-level controlled the downslope transport of sediment from the Maldives archipelago to the deep-sea environments and influenced the current strength at the benthic boundary layer of the Inner Sea resulting in different grain sizes and substrates. The drift deposits recovered by core M74/4-1143 have shown that highest current intensities occurred during and after the glacial terminations (Fig. 6). Bottom currents in general were stronger during interglacials than during glacials, although core SO236-052-4 show lower current velocities and a lower amplitude of change. Stronger current intensities at the sea floor likely favoured the distribution of certain suspension feeding benthic foraminiferal taxa, such as *D. araucana*.

(4) The long-term trend in the benthic $\delta^{13}$C record mirrors the basin-wide change in the composition of intermediate waters, implying a close linkage to the main formation sites of the AAIW in the Southern Ocean. The precessional changes of estimated oxygen concentrations of intermediate waters are coherent with changes in the deep Arabian Sea. This suggests an influence of the lateral expansion of oxygen minimum waters from the Arabian Sea into the equatorial intermediate Indian Ocean and modulation by inflowing AAIW from the south. The predominance of *N. proboscidea* during a long phase of reduced oxygen concentrations (with average oxygen concentrations around 50 µmol kg$^{-1}$) during late MIS 5 to late MIS 3 suggests an adaption of this species to the particular biogeochemical conditions and food quality associated with low oxic conditions.

**Acknowledgements**

We thank the masters and crews as well as the shipboard scientific parties of R/V SONNE SO236 and R/V METEOR M74/4 cruises for their excellent collaboration. Jutta Richarz is thanked for her support during grain size analyses, and Lisa Schönborn and Günther Meyer for technical support during stable isotope measurements. Aurora Elmore and Thomas Ronge are thanked for providing the stabile carbon isotope records of the southwestern Pacific Ocean (cores DSDP593, MD06-2986, MD06-2990/SO136/003) and we thank Yvonne Milker for the many helpful discussions. We acknowledge Luc Beaufort for editorial comments and the two anonymous reviewer, who helped to improve this paper with detailed comments and suggestions considerably. This research used data acquired at the XRF Core Scanner Lab at the MARUM, Center for Marine Environmental Sciences, University of Bremen, Germany. This study was supported by grants 03G0236A of the Federal Ministry of Education and Research. The Ministry of Fisheries and Agriculture of the Maldives is thanked for granting the research permit for Maldivian waters.

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

**Table Captions**

**Table 1:** Accelerator Mass Spectrometry (AMS) radiocarbon dating results based on mixed surface-dwelling planktonic foraminifera (*Gr = Globigerinoides ruber*, white; *Gs = Globigerinoides sacculifer*) from 35 cm, 80 cm and 140 cm sediment depth of core SO236-052-4. Conventional radiocarbon ages were calibrated using the radiocarbon calibration program CALIB (version 7.0.4; Stuiver and Reimer, 1993) and the calibration curve Marine13 (Reimer et al., 2013).

| No. | Sample ID | Lab ID | Material | Core depth | $^{12}C/$ $^{13}C$ | $^{14}C$ age | Calibrated age (ΔR 0) | | |
|---|---|---|---|---|---|---|---|---|---|
| | | | | | | | cal BP (2s ranges, 95.4 % probability) | | |
| | | | | [mbsf] | o/oo | ya BP | range [years] | rel. area u. distr. | median of prob. [ka] |
| **1** | SO236-052-035 | Beta-418574 | *Gr, Gs* | 0.35 | +1.4 | 7940 ±30 | 8330 - 8480 | 1.00 | **8.4 ± 0.08** |
| **2** | SO236-052-080 | Beta-418575 | *Gr, Gs* | 0.80 | +1.6 | 12890 ±40 | 14310 - 15020 | 1.00 | **14.7 ± 0.36** |
| **3** | SO236-052-140 | Beta-418576 | *Gr, Gs* | 1.40 | +1.8 | 23930 ±100 | 27480 - 27850 | 1.00 | **27.7 ± 0.19** |

**Table 2:** Species composition of benthic foraminiferal assemblages. Principal component number, dominant and important associated species with principal component scores (Q-mode) and explained variance in percent of total variance are given.

| Q-mode Principal Components | Species | Scores | Explained variance [%] |
|---|---|---|---|
| PC1 | ***Neouvigerina proboscidea*** | 5.812 | 31.54 |
| | ***Discorbinella araucana*** | 3.948 | |
| | *Hyalinea inflata* | 2.562 | |
| | *Cymbaloporetta squammosa* | 1.913 | |
| | *Bulimina marginata* | 1.729 | |
| | *Rosalina vilardeboana* | 1.595 | |
| PC2 | ***Cibicides mabahethi*** | 7.466 | 30.54 |
| | *Discorbinella bertheloti* | 1.756 | |
| | *Siphogenerina columellaris* | 1.622 | |
| | *Gyroidina umbonata* | 1.589 | |
| | *Reophax* sp. | 1.387 | |
| | *Hyalinea inflata* | 1.214 | |
| | *Discorbinella araucana* | 1.109 | |
| PC3 | ***Neouvigerina proboscidea*** | 4.608 | 27.06 |
| | ***Hoeglundina elegans*** | 3.952 | |
| | *Discorbinella bertheloti* | 3.004 | |
| | *Cibicidoides subhaidingeri* | 2.311 | |
| | *Discorbis* sp. | 2.161 | |
| | *Spiroplectinella sagittula* s.l. | 1,808 | |
| | *Cibicides mabahethi* | 1,084 | |

**Figure Captions**

**Figure 1:** Location maps of the Maldives archipelago in the Indian Ocean (a, b) and the setting of the study area (c) (modified after Betzler et al., 2013a), showing the location of sediment core M74/4-1143 in the Kardiva Channel and core SO236-052-4 in the central part of the Inner Sea (red circles).


**Figure 2:** Full resolution stable oxygen isotope records of the planktonic foraminifer *G. ruber* (a) and age-depth plots for the sediment cores SO236-052-4 (light blue) and M74/4-1143 (grey). Orange triangles indicate radiocarbon dates and circles indicate age control points derived from correlation with the LR04 benthic isotope stack of Lisiecki and Raymo (2005). Sedimentation rates are derived from linear interpolation between age data. MIS denotes the Marine stable oxygen isotope

stages.

**Figure 3:** a-b) Stable oxygen and carbon isotope records of planktonic and benthic foraminifera of sediment core SO236-052-4. Displayed are the planktonic species *G. ruber* (light blue), the epibenthic species *C. mabahethi* (dark blue) and the deep infaunal species *G. affinis* s.l. (red). Also shown are the full-resolution data of the $\Delta\delta^{13}C_{Gr-Cm}$ and $\Delta\delta^{13}C_{Cm-Ga}$ signals

(c), as a result of the difference between the $\Delta\delta^{13}C$ values of *G. ruber* and *C. mabahethi*, as well as *C. mabahethi* and *G. affinis* s.l. MIS denotes the Marine stable oxygen isotope stages.

**Figure 4:** Records and normalized Blackman-Tukey power spectra of a, b) the oxygen reconstructions of core SO236-052-4 (blue) and GeoB3004 (purple), c, d) productivity proxies TOC (green) and Br XRF counts (pink), and e, f) dust proxies Ti/Al

(light blue) and Fe/Al (dark blue), in comparison to the Δ-insolation at 30° N (black dashed line). Grey bars indicate the 23 ka period of the Δ-insolation and its band width. Oxygen and TOC records represent five-point running averages, Br XRF counts, Ti/Al and Fe/Al records represent fifteen-point running averages.

**Figure 5:** Sedimentological and geochemical records of sediment core SO236-052-4 from the central part of the Maldives

Inner Sea. a) Sortable silt (black) in comparison with the data of core M74/4-1143 (brown) and bulk sediment (grey) MEAN values, b) total organic carbon (TOC) (dark green) and calcium carbonate (light green) content of the sediment, c) iron (dark blue) and titanium (light blue) in relation to the aluminium XRF counts, and d) bromine XRF counts (pink), and silicon (dark purple) and strontium (light purple) in relation to the calcium XRF counts. Thin lines represent full-resolution data, bold lines in a) and b) indicate five-point running averages, whereas all XRF counts in c) and d) are displayed as a fifteen-point

running average (bold lines). MIS denotes the Marine stable oxygen isotope stages.

**Figure 6:** a) Epibenthic stable oxygen isotope record of core SO236-052-4 (dark blue) in comparison with the LR04 benthic stable isotope stack (orange; Lisiecki and Raymo, 2005). b-c) Comparison of the sortable silt records of sediment cores M74/4-1143 and SO236-052-4, and d) relative abundance of meroplanktonic benthic foraminifera (BF; including the genera

*Cymbaloporetta* and *Tretomphaloides*) in sediments of core SO236-052-4. Thin lines represent full-resolution data, bold lines indicate five-point running averages. MIS denotes the Marine stable oxygen isotope stages.

**Figure 7:** Comparison of benthic foraminiferal faunal records of core SO236-052-4 from the central part of the Maldives Inner Sea. a) Shannon-Wiener diversity index H(S), b-d) Q-mode benthic foraminiferal assemblages, including the *N.*

*proboscidea-/D. araucana*-fauna (assemblage 1), the *C. mabahethi*-fauna (assemblage 2), and the *N. proboscidea-/H. elegans*-fauna (assemblage 3). Loadings ≥ 0.5 are defined as significant after Backhaus et al. (2008). e-g) Distribution of selected important and associated benthic foraminiferal taxa, given in percent. The meroplanktonic benthic foraminifera (BF) comprise the genera *Cymbaloporetta* and *Tretomphaloides*.

 **Figure 8:** Variation of the insolation difference between the June and December solstice at 30° N (after Laskar, 2004; calculated with AnalySeries 2.0: Paillard et al. 1996) (a) in graphical correlation with geochemical and benthic foraminiferal productivity records of core SO236-052-4. b) Total organic carbon (TOC) content and c) Bromine as derived from XRF scanning count data as tracer for marine organic carbon (MOC), d) Fe/Al ratio and e) relative abundance of agglutinated benthic foraminifera as indicator for enhanced dust supply. f) Principal Components (PC) show the *C. mabahethi*-fauna (assemblage 2) and *N. proboscidea*-/*H. elegans*-fauna (assemblage 3). This comparison reveals coherent glacial-to-interglacial changes in all proxy records with elevated values during glacial stages MIS 6 and MIS 4-2 and during a weaker northern hemisphere solar radiation-amplitude (yellow bars). Thin lines represent full-resolution data, bold lines in b) and e) indicate five-point running averages, bold lines in c) and d) indicate a fifteen-point running average. MIS denotes the Marine stable oxygen isotope stages.

**Figure 9:** Water mass circulation changes obtained from stable $\delta^{13}$C data of core SO236-052-4 (Indian Ocean) in comparison to ventilation changes in the Arabian Sea. a) $\delta^{13}$C records of the planktonic *G. ruber* (light blue) and the epibenthic *C. mabahethi* (dark blue), b) difference between the planktonic and epibenthic stabile carbon records ($\Delta\delta^{13}C_{Gr-Cm}$), c) differences between epibenthic and deep endobenthic $\delta^{13}$C records of SO236-052-4 (dark blue) from the intermediate Maldives Inner Sea in comparison to that of GeoB3004 (purple) from the deep Arabian Sea (Schmiedl and Mackensen, 2006). Changes in intermediate- and deep-water oxygen concentrations are calculated by the linear regression between the $\Delta\delta^{13}$C and $[O_2]$ < 235 µmol kg$^{-1}$ after Hoogakker et al. (2015). Minimum significance level of $[O_2]$ = 55 µmol kg$^{-1}$ is shown as dashed line (Hoogakker et al., 2015). Variation of the insolation difference between the June and December solstice at 30° N (yellow) were estimated after Laskar (2004) with AnalySeries 2.0 (Paillard et al., 1996). All lines indicate five-point running averages. MIS denotes the Marine stable oxygen isotope stages. *Cib.* = *Cibicides*, *Cm* = *Cibicides mabahethi*, *Cw* = *Cibicides wuellerstorfi*, *Ga* = *Globobulimina affinis*, *Gr* = *Globigerinoides ruber*.

**Figure 10:** Comparison of epibenthic $\delta^{13}$C records from intermediate water depth of the equatorial Indian and temperate southwestern Pacific Ocean. a) $\delta^{13}$C record of *C. mabahethi* (blue) from the Maldives Inner Sea (SO236-052) in comparison to $\delta^{13}$C records from the southwestern Pacific Ocean, which were mainly generated from *Cibicidoides wuellerstorfi*, *C. cicatricosa* and *C. kullenbergi* and represent different water depths (Thiede et al., 1999; Pahnke and Zahn, 2005; Elmore et al., 2015; Ronge et al., 2015). Data of MD97-2120 were traced after Pahnke and Zahn (2005). All lines represent five-point running averages. b) Simplified map of the Indian Ocean and the southwestern Pacific Ocean with location of the study sites.

990     Figure 1

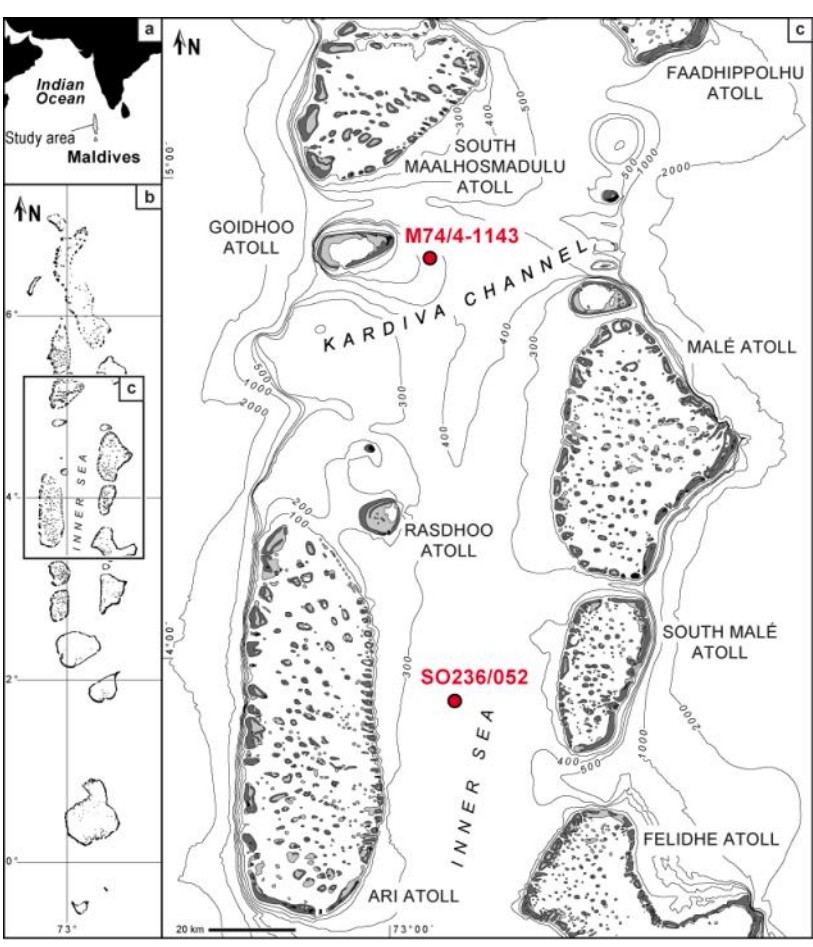

Figure 2

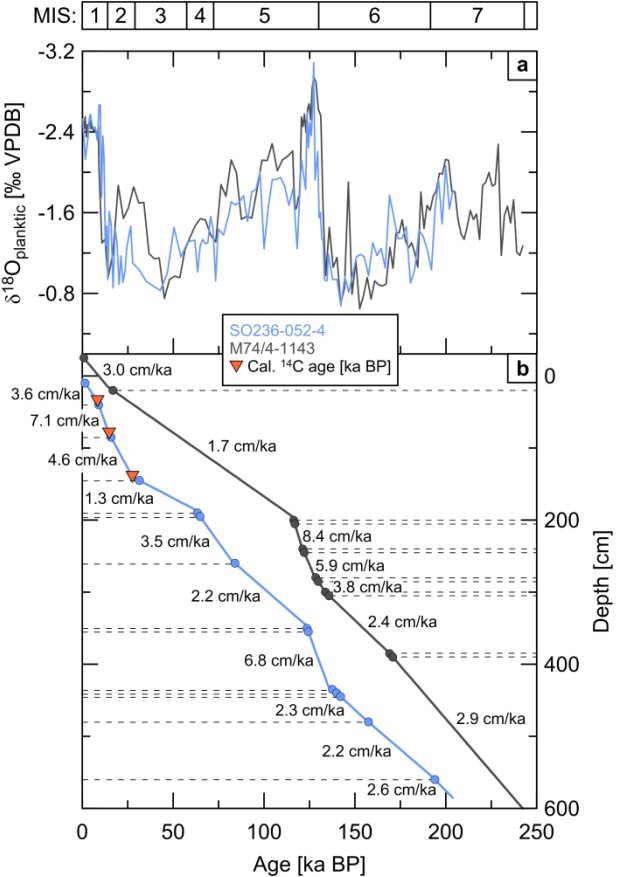

Figure 3

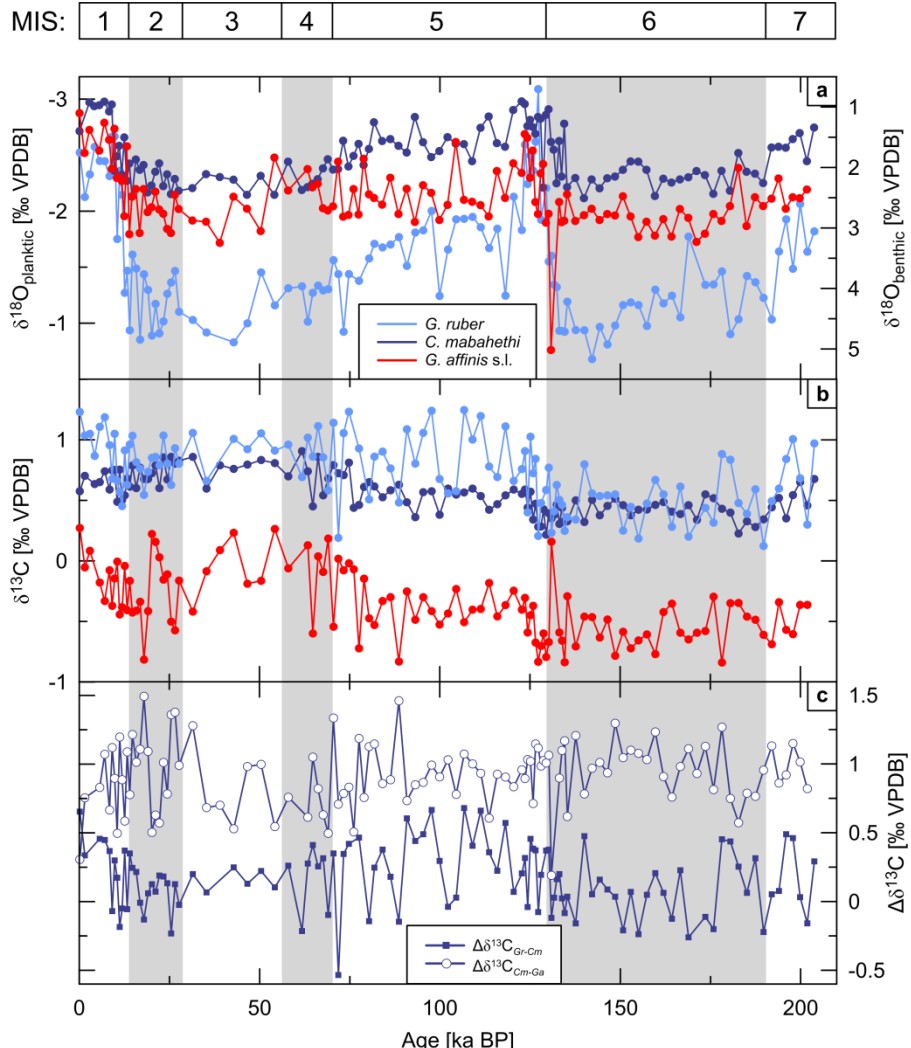

Figure 4

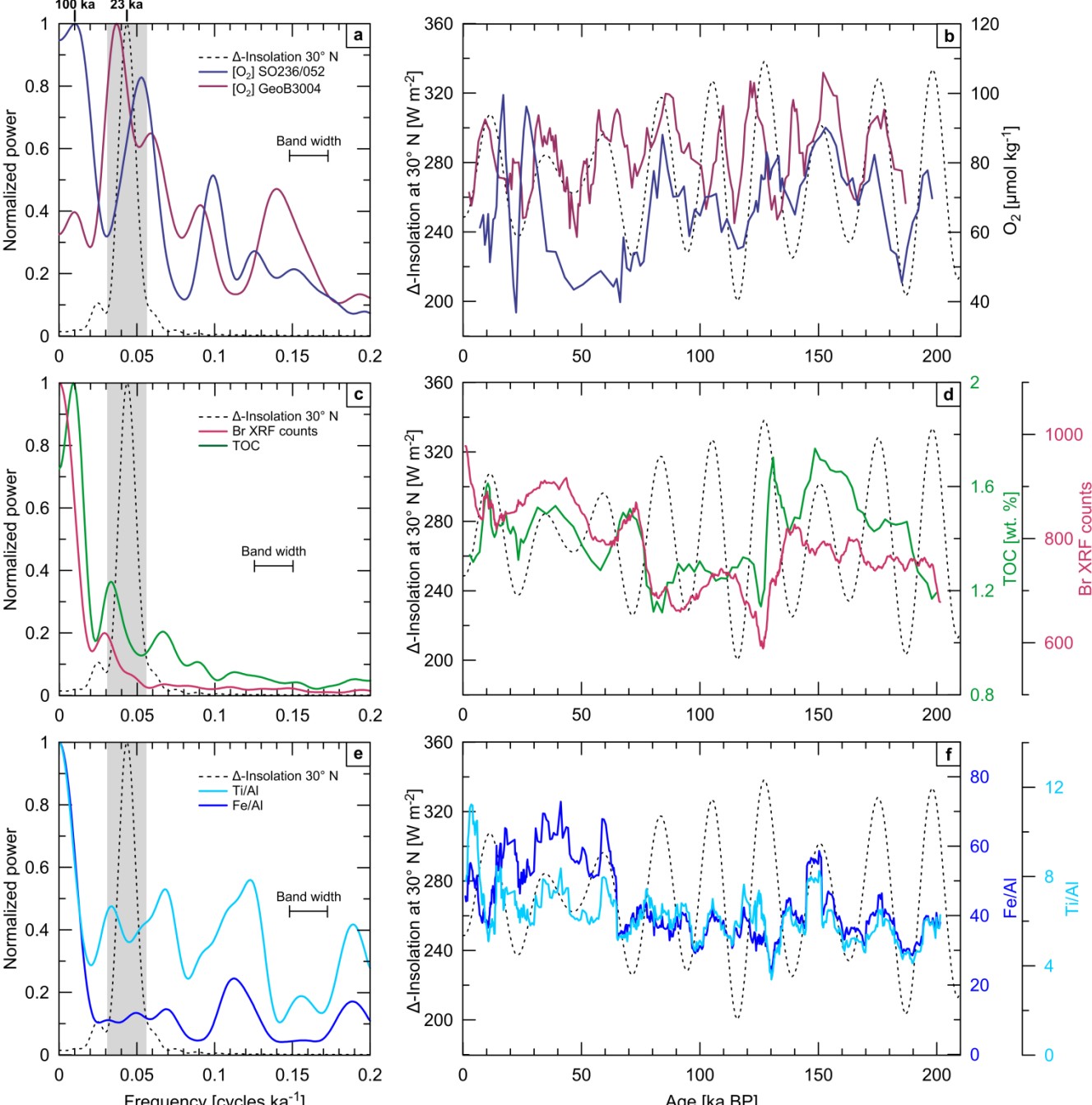

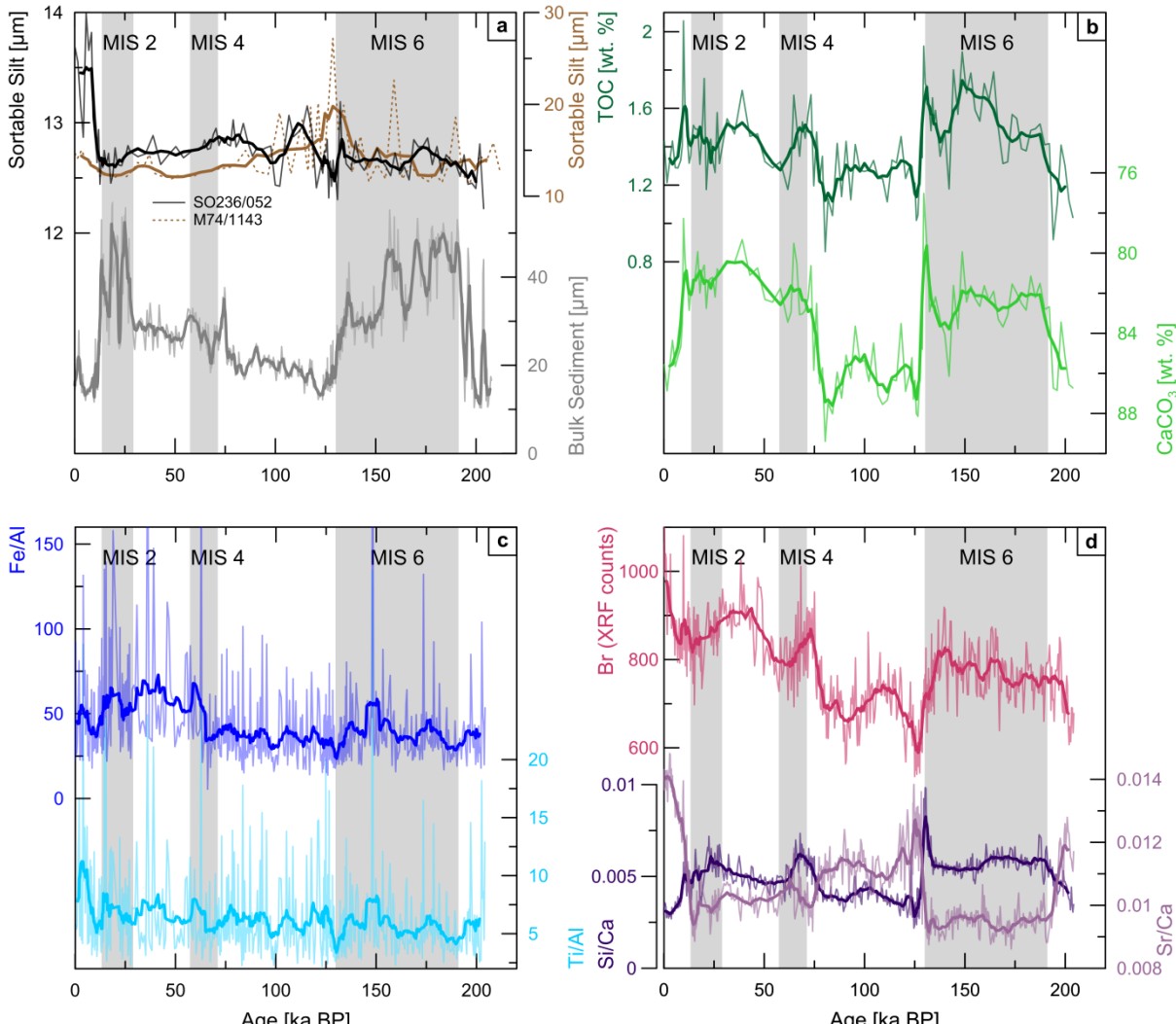

Figure 6

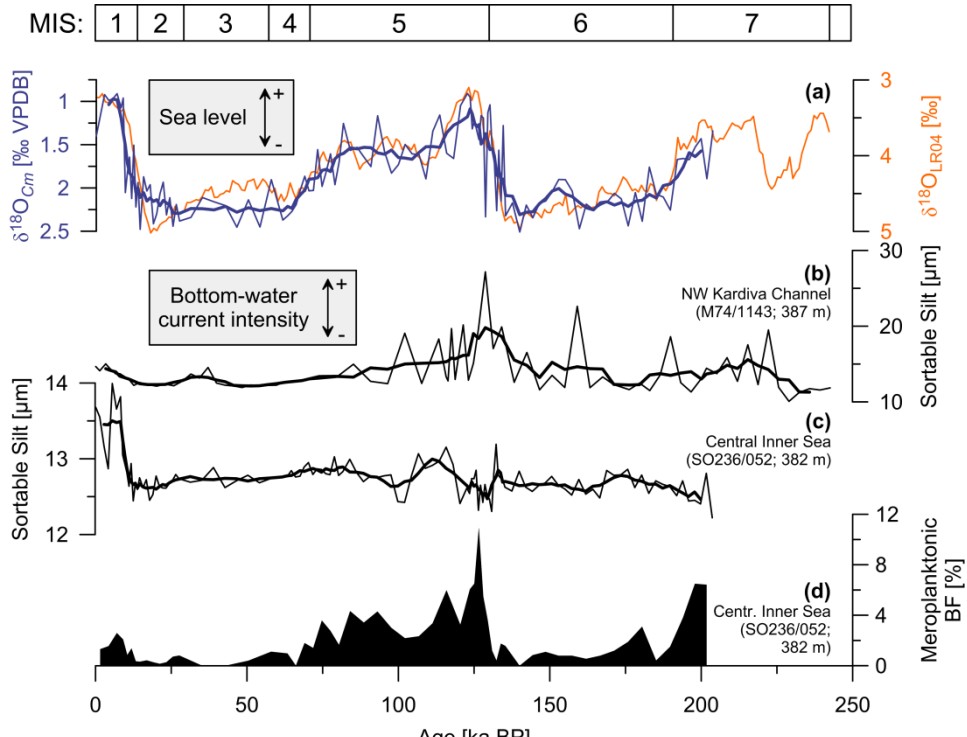

Figure 7

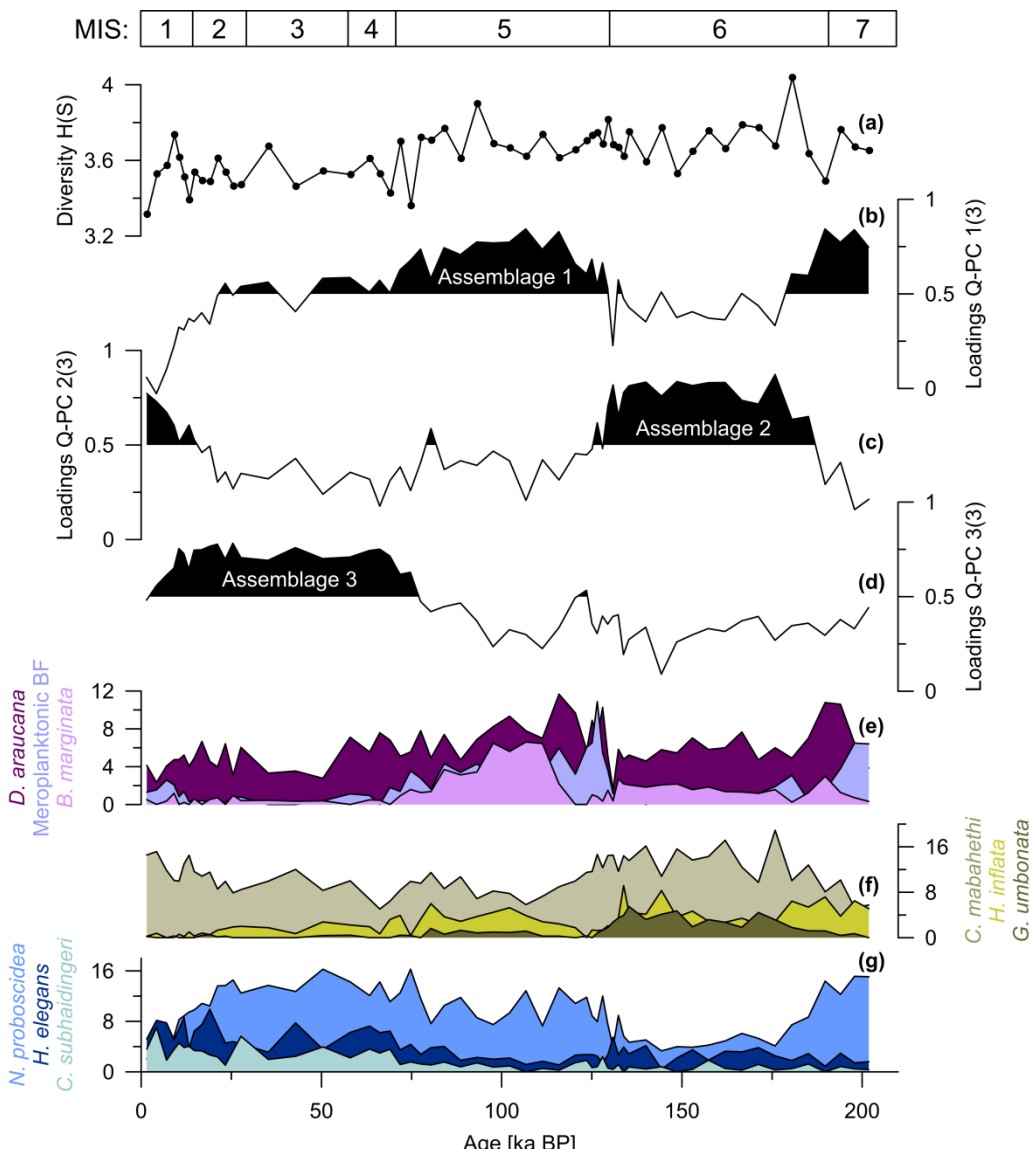

Figure 8

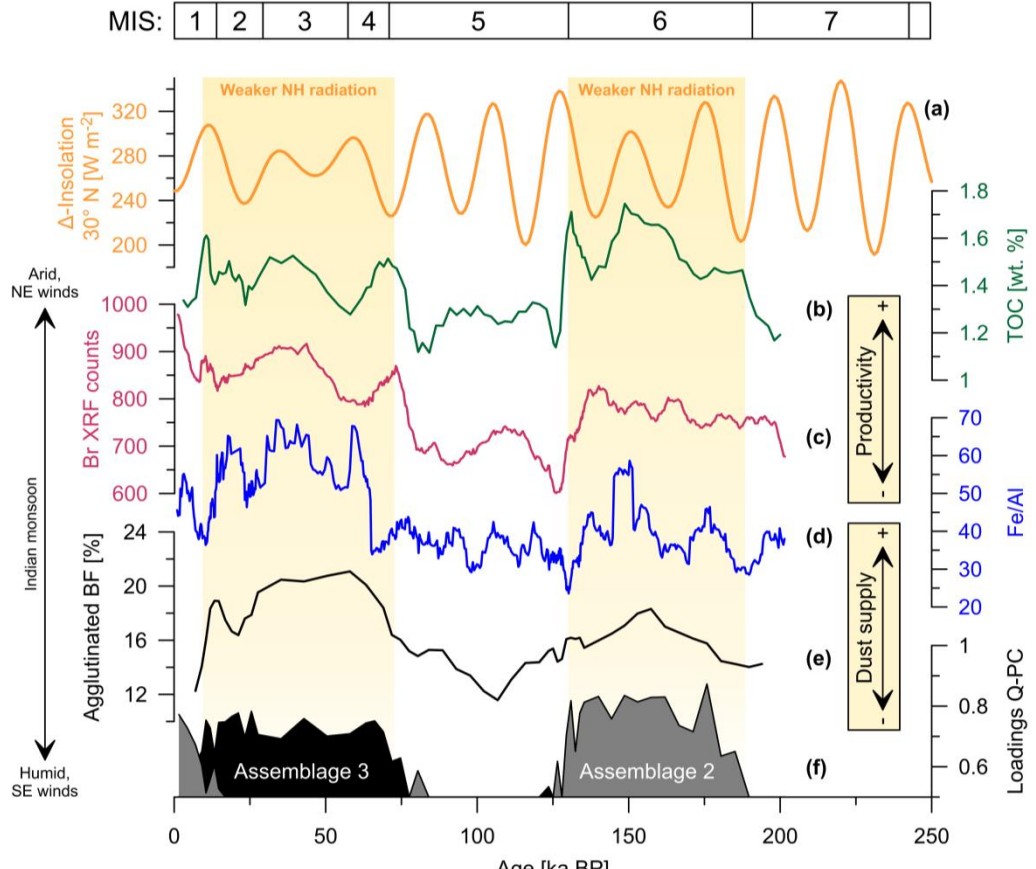

Figure 9

1000

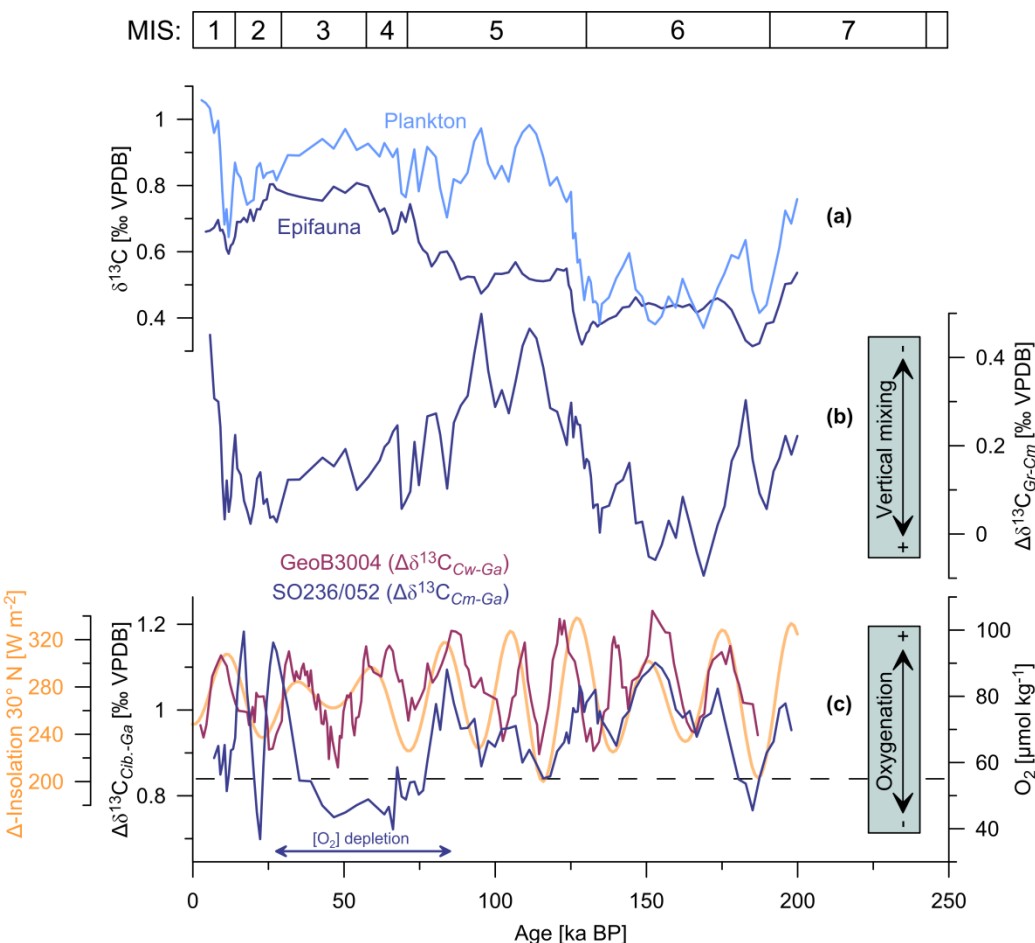

Figure 10

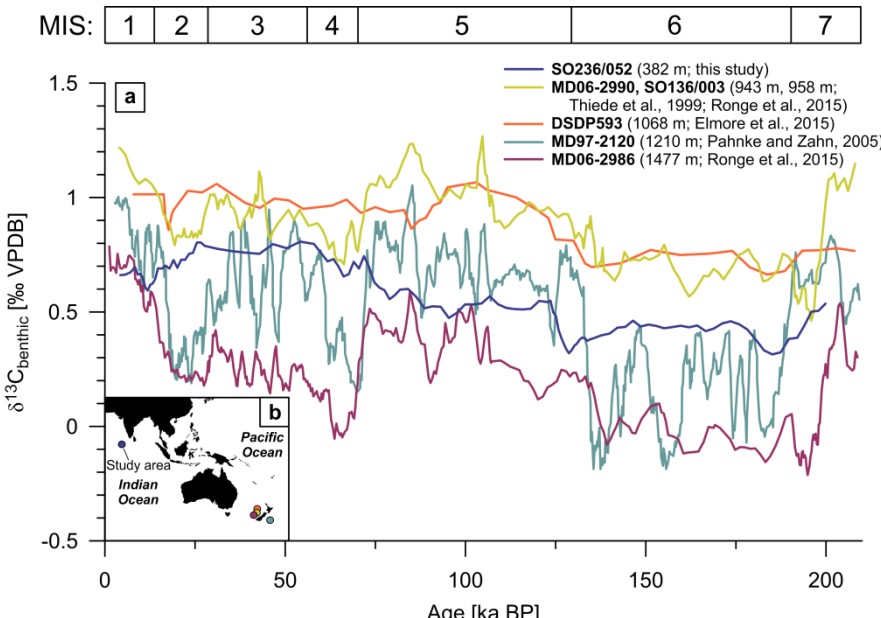