# Peer review of "A multi-proxy analysis of Late Quaternary ocean and climate variability for the Maldives, Inner Sea"

_Climate of the Past, 2017_

## Referee Comment (RC1) · Anonymous Referee #1 · 16 May 2017

Bunzel et al., present a multi-proxy data record from the Maldives (equatorial Indian Ocean) over the last 200 kyr. The integrated evaluation of proxy records suggest a close linkage between the Indian monsoon oscillation, intermediate water circulation, productivity and sea-level changes on orbital time-scale in the Maldives Inner sea.

General comments: The paper of Bunzel et al., is an interesting contribution to our understanding of the climate of the equatorial Indian Ocean. I have two main concerns that I would like to see addressed before the paper can be published in Climate of the Past. First, the authors often refer to previous published works in their manuscript but the data are not presented on the Figures of the paper, making the comparison with these previous works very difficult. Second, the authors always discuss the periodicity in their different proxies or the timing of variability (example: variability in the precession band, phases of reduced northern hemisphere insolation. . ..) but there is no statistics

to confirm the significance of the periodicities and the exact timing (spectral analyses) that are discuss in the manuscript. This also hamper to compare with the variability found in other published records. I therefore recommend major revision for the current manuscript.

Below are my specific comments: 1) I don't find the title of the current manuscript really suitable. The title suggest that the Maldives record is mainly driven by "Indian monsoon dynamics" whereas the authors conclude that the record provide a close linkage between the Indian monsoon oscillation, intermediate water circulation, productivity and sea-level changes on orbital time-scale. Therefore, a title such as "A multi-proxy analysis of late Quaternary equatorial Indian ocean for the Maldives, Inner Sea" could be less confusing.

2) Lines 57 to 59. There is much more references of Arabian Sea works at the orbital and suborbital time scales (Clemens et al., 1996; Altabet et al., 2002; Clemens and Prell, 2003; Pichevin et al., 2007; Boning and Bard, 2009; Ziegler et al., 2010; Caley et al., 2011; Caley et al., 2013, Deplazes et al., 2013 are some examples).

3) Lines 179-180: "Local reservoir corrections were not applied". The authors should explain why they do not applied a correction. In general a correction of 400 years is applied in the tropics.

4) Lines 202-209: Oxygen concentration should be shown on figure 3 and not only on Figure 8.

5) Lines 217-221: The data of core M74/4-1143 are not shown on figure 4 making the comparison with core SO-236-052-4 impossible.

6) Lines 228-233 and 258-259; Are the Fe/Ca and Si/Ca good proxies for Aeolian dust? The results could be compared to dust data from site ODP722 (Clemens et al., 1996). This is important to discuss the provenance of the dust and the interpretation of the Fe/Ca and Si/Ca that stays speculative in the discussion (lines 258-268). Also, previous

study in the Arabian Sea used rather the changes in the Ti/Al ratio of the sediments as indicator for grain size and thus wind speed, since Titanium is concentrated in heavy minerals in the coarser size fraction (Reichart et al., 1997; Ziegler et al., 2010 CP).

7) Line 278: "Fe/Ca record lacks significant variability on the precession band". Statistical analyses are necessary (spectral analyses) to confirm this point.

8) Lines 336-337: "While the benthic foraminiferal fauna preliminary show changes on glacial-interglacial time scale, the TOC content and Ba/Ca ratio are characterized by additional variability in the precessional band." Again, statistical analyses are necessary (spectral analyses) to confirm this point.

9) Lines 342-347: "Elevated TOC and Ba/Ca ratios at site SO-236-052 during phases of reduced northern hemisphere summer insolation suggest a direct influence of the Indian winter monsoon on productivity and related organic matter fluxes of the Maldives Inner Sea during the past 200 ka, which is consistent with the present-day situation (de Vos et al. 2014). The close link between the winter monsoon intensity and surface water productivity in the study area is confirmed by the difference between the $\delta$13C values of the epipelagic G. ruber (Gr) and the epibenthic C. mabahethi (Cm) (Figs. 3, 8)." Again, statistical analyses are necessary with a phase analyse. Also, what could be the role of the IEW and ENSO mentioned in the introduction part?

10) Lines 372-376: "term trends of similar magnitude have been recorded from sites bathed by the Antarctic Intermediate Water mass (AAIW) in the southwestern Pacific Ocean (Pahnke and Zahn, 2005; Elmore et al., 2015; Ronge et al., 2015). The general resemblance of the various epibenthic $\delta$13C records suggests a 375 significant role of AAIW in ventilation of bathyal environments of the Maldives Inner Sea, which is consistent with the modern oceanographic situation (You, 1998)." It would be good to add the data of the previous work mentioned on the Figure of the manuscript for a direct comparison.

11) Lines 385-386: "The reconstructed O2 record reveals precessional changes between oxic and low oxic conditions during northern hemisphere insolation maxima and minima, respectively". Statistical analyses are necessary (spectral analyses) to confirm this point.

12) Lines 406-412: "Agulhas leakage". I do not understand this paragraph and the link with the Agulhas leakage. If the authors want to demonstrate a link between their record and the Indian monsoon they can compare directly with published monsoon records. Also, the forcing of the Agulhas leakage at terminations is driven by the sub-tropical front migration and is not directly link to the Indian monsoon (Peeters et al., 2004). For the IEW impact, a statistical analyze with the phase relationship (spectral analyses) will help the interpretation of the record.

References

Altabet, M. A., Higginson, M. J., & Murray, D. W. (2002). The effect of millennial-scale changes in Arabian Sea denitrification on atmospheric CO2. Nature, 415(6868), 159-162. Böning, P., & Bard, E. (2009). Millennial/centennial-scale thermocline ventilation changes in the Indian Ocean as reflected by aragonite preservation and geochemical variations in Arabian Sea sediments. Geochimica et Cosmochimica Acta, 73(22), 6771-6788. Caley, T., Malaizé, B., Zaragosi, S., Rossignol, L., Bourget, J., Eynaud, F., ... & Ellouz-Zimmermann, N. (2011). New Arabian Sea records help decipher orbital timing of Indo-Asian monsoon. Earth and Planetary Science Letters, 308(3), 433-444. Caley, T., Zaragosi, S., Bourget, J., Martinez, P., Malaizé, B., Eynaud, F., ... & Ellouz-Zimmermann, N. (2013). Southern Hemisphere imprint for Indo-Asian summer monsoons during the last glacial period as revealed by Arabian Sea productivity records. Biogeosciences, 10(11), 7347. Clemens, S. C., Murray, D. W., & Prell, W. L. (1996). Nonstationary phase of the Plio-Pleistocene Asian monsoon. Science, 274(5289), 943. Clemens, S. C., & Prell, W. L. (2003). A 350,000 year summer-monsoon multi-proxy stack from the Owen Ridge, Northern Arabian Sea. Marine Geology, 201(1), 35-51. Deplazes, G., Lückge, A., Peterson, L. C., Timmermann, A., Hamann, Y., Hughen, K. A., ... & Haug, G. H. (2013). Links between tropical rainfall and North Atlantic climate

during the last glacial period. Nature Geoscience, 6(3), 213-217. Peeters, F. J., Acheson, R., Brummer, G. J. A., De Ruijter, W. P., Schneider, R. R., Ganssen, G. M., ... & Kroon, D. (2004). Vigorous exchange between the Indian and Atlantic oceans at the end of the past five glacial periods. Nature, 430(7000), 661-665. Pichevin, L., Bard, E., Martinez, P., & Billy, I. (2007). Evidence of ventilation changes in the Arabian Sea during the late Quaternary: Implication for denitrification and nitrous oxide emission. Global Biogeochemical Cycles, 21(4). Reichart, G. J., den Dulk, M., Visser, H. J., van der Weijden, C. H., & Zachariasse, W. J. (1997). A 225 kyr record of dust supply, paleoproductivity and the oxygen minimum zone from the Murray Ridge (northern Arabian Sea). Palaeogeography, Palaeoclimatology, Palaeoecology, 134(1-4), 149-169. Ziegler, M., Lourens, L. J., Tuenter, E., Hilgen, F., Reichart, G. J., & Weber, N. (2010). Precession phasing offset between Indian summer monsoon and Arabian Sea productivity linked to changes in Atlantic overturning circulation. Paleoceanography, 25(3). Ziegler, M., Lourens, L. J., Tuenter, E., & Reichart, G. J. (2010). High Arabian Sea productivity conditions during MIS 13–odd monsoon event or intensified overturning circulation at the end of the Mid-Pleistocene transition?. Climate of the Past, 6(1), 63-76.

---

## Referee Comment (RC2) · Anonymous Referee #2 · 17 May 2017

I am reviewing here a manuscript by Bundel et al. on a multi-proxy analysis of a sediment core from the Maldives region. The data is interpreted in context of dust input, sea-level changes and oceanographic changes over the last 200.000 years.

Overall I think that this study and data can be eventually suitable for publication in Climate of the Past. It touches upon topics that have been published previously in the journal and the datasets will be of interest to the paleoceanographic community that works in the Indian Ocean during the Late Pleistocene.

Overall, I think the analysis of the data and its presentation could be improved. Some of the statements in the discussion (e.g. correlation of certain proxies with insolation or other proxy records, glacial-interglacial cycles) is often not supported by statistics or suitable figures (see comments below).

[Figure]

I am also missing clear common thread and objective. This starts already in the abstract. It's starts of with a paragraph that basically says : "We measured a lot of stuff on a sediment core in the Maldives region. . .and then we interpreted the data..".

I think it would be much more appealing if the manuscript would start with the context of the study, the main research question or problem or a hypothesis. Then they should list their approach (multi-proxy approach)

Further comments:

I think the paragraph from line 57 to line 66 could be improved. This paragraph contains a controversy in the interpretation of past OMZ variability in the Arabian Sea and its relation with summer monsoon variability. Is a strong OMZ linked to increased productivity (monsoon driven upwelling) or reduced ventilation (lower oxygen conc in southern sourced intermediate waters). The study by Bundel et al could inform this debate by providing a record of oxygen concentration from further South. There are some records (e.g. Ziegler et al., 2010, Climate of the Past) that show that a deep (most extended) OMZ occurs during glacial periods. While productivity maxima in the Arabian Sea, occur during interglacials. The new data by Bundel et al., could help to explain this observation by proving constraints of the Arabian Sea intermediate water ventilation from the South.

Line 64: studies in stead of studied?

line 77 -82: This section lists the main objectives of the study. Its strange that the objectives 1 and 2 mention suddenly, dust flux and sea-level, while the entire introduction does not mention either of the two. I would focus on objective 3 and mention the subjects that deal with 1 and 2 in the discussion without putting to much emphasis on it.

line170: ". . .are based on. . ."

line 173: 'was estimated to assess.."

line 179-180: Why was a local reservoir age not applied?

line 224: Given that the authors did XRF scanning, they should also have Bromine data. Bromine has been used successfully in several studies in the Indian Ocean as organic matter indicator (Caley et al., 2013, QSR, Ziegler et al, 2008, G3). The authors could do the same to get a high-resolution organic matter record and get a better idea of short term variability in TOC.

line 258-260: What about the possibility that Fe/Ca and Si/Ca reflect changes in carbonate production / preservation? Maybe the dust input has been constant through time? See also related comments by the other reviewer. I fully agree with him.

line 278: at the precessional band

line 282: There are several studies that suggest that late Pleistocene quasi-100 kyr cycles are not driven by eccentricity, but instead are a response to skipped precession and/or obliquity cycles

line 342-345: I don't see a correlation of TOC or Ba with summer insolation. This should be demonstrated in a figure.

section 4.2: This section seems not to be very important in the context of the whole manuscript. I would therefore again suggest to omit the sea-level topic from the list of main objectives.

line 370-376: show the comparison with other datasets also in the figures otherwise the reader cannot judge your arguments

line 388-390: This sentence seems to contain a contradiction. Is the Maldives OMZ controlled by expansion of the Arabian Sea OMZ are controlled by the ventilation of southern sourced waters. (I would think it is the latter)

line 396-401: I would argue the other way around. Low oxygen conc in intermediate waters in the Maldives area preconditioned the waters that ventilate the Arabian Sea.

[Figure]

So a deep Arabian Sea OMZ has its root in the central Indian Ocean (and is thus not exclusively controlled by monsoon variability).

line 428: demonstrate cyclicality through spectral analysis (see also comment by other reviewer, fully agree)

Figure 6: Why is assemblage 2 abundant in the glacial MIS 6 and the Holocene? (Why is assemblage 1 abundant in 5, but absent in the Holocene)?

---

## Author Comment (AC1) · 23 Jun 2017

Response to Referee#1,

We acknowledge the substantial comments by the reviewer, and especially for bringing up the suggestion of including a more profound statistical evaluation of our data series, the usage of alternative element ratios as dust indicators, and the suggestion for comparison of stable carbon isotope records from various intermediate water sites. Specifically, we have generated Blackman-Tukey power spectra for the most relevant proxy records supporting our paleoenvironmental interpretation. The comments helped to improve our manuscript considerably. Below we respond to all comments raised by the reviewer. With kind regards, Dorothea Bunzel

1) I don't find the title of the current manuscript really suitable. The title suggest that the Maldives record is mainly driven by "Indian monsoon dynamics" whereas the authors conclude that the record provide a close linkage between the Indian monsoon oscillation, intermediate water circulation, productivity and sea-level changes on orbital time-scale. Therefore, a title such as "A multi-proxy analysis of late Quaternary equatorial Indian ocean for the Maldives, Inner Sea" could be less confusing.

Response: Thank you for suggesting a more suitable title. We will adjust the title accordingly, e.g. "A multi-proxy analysis of late Quaternary ocean and climate variability for the Maldives, Inner Sea"

2) Lines 57 to 59. There is much more references of Arabian Sea works at the orbital and suborbital time scales (Clemens et al., 1996; Altabet et al., 2002; Clemens and Prell, 2003; Pichevin et al., 2007; Boning and Bard, 2009; Ziegler et al., 2010; Caley et al., 2011; Caley et al., 2013, Deplazes et al., 2013 are some examples).

Response: We agree and will include the mentioned references in the revised version.

3) Lines 179-180: "Local reservoir corrections were not applied". The authors should explain why they do not applied a correction. In general a correction of 400 years is applied in the tropics.

Response: We did not correct our radiocarbon ages for local reservoir effects, because the closest available numbers of marine reservoir age corrections are from the Arabian Sea, Northern Indian Ocean and Bay of Bengal, between 821 and 864 km distance from our study site. These reservoir age correction values vary between 301 to 544 years (Dutta et al., 2001; Southon et al., 2002). Due to the contrasting values and in order to minimize potential errors we decided to apply the global marine reservoir correction of 400 yrs. We will change the text accordingly (lines 178-179): "The AMS 14C ages were corrected for the global reservoir age of 400 years and converted to calendar years using the radiocarbon calibration program CALIB (version 7.0.4; Stuiver and Reimer, 1993)."

4) Lines 202-209: Oxygen concentration should be shown on figure 3 and not only on Figure 8.

Response: We will include the $\Delta\delta13CCm$-Ga data, which were used for the bottom-water oxygen reconstruction, in Figure 3.

5) Lines 217-221: The data of core M74/4-1143 are not shown on figure 4 making the comparison with core SO-236-052-4 impossible.

Response: We will include the sortable silt data of core M74/4-1143 in Figure 4a.

6) Lines 228-233 and 258-259; Are the Fe/Ca and Si/Ca good proxies for Aeolian dust? The results could be compared to dust data from site ODP722 (Clemens et al., 1996). This is important to discuss the provenance of the dust and the interpretation of the Fe/Ca and Si/Ca that stays speculative in the discussion (lines 258-268). Also, previous study in the Arabian Sea used rather the changes in the Ti/Al ratio of the sediments as indicator for grain size and thus wind speed, since Titanium is concentrated in heavy minerals in the coarser size fraction (Reichart et al., 1997; Ziegler et al., 2010 CP).

Response: We consider the more commonly used proxies for terrigenous sediment delivery/aeolian dust fluxes and we will modify our manuscript as follows: We will replace the Fe/Ca record by the Ti/Al and Fe/Al records as proxies for aeolian dust supply and enhanced aridity of the hinterland/source area (e.g. Zhang et al., 1993; Lourens et al., 2001; Itambi et al., 2009) in order to account for a potential influence of changes in carbonate production and preservation on the Fe/Ca ratio (Wehausen and Brumsack, 2000). We have included the Fe/Al record because the aeolian Fe flux to the ocean likely has a direct impact on the seasonal surface ocean productivity (e.g. Martin et al., 1991; Boyd et al., 2000; Gao et al., 2001; Jickells et al., 2005), which also influences the deep-sea benthic ecosystems through seasonal phytodetritus pulses. Both Ti/Al and Fe/Al records show a similar glacial-interglacial pattern with relatively higher values during cold stages corroborating the foraminiferal results of enhanced surface ocean fertilisation of the Maldives Inner Sea during glacial periods. We also show and

discuss the Si/Ca ratio in addition, because we argue that the increase in agglutinated benthic foraminifera is a reflection of the availability of terrestrial particles, since most of these agglutinated species preferentially use siliciclastic grains for building up their test walls (e.g., Murray, 2006). We will refer to the lithogenic flux record of site ODP722 (Clemens et al., 1996) since it supports our observation of generally enhanced glacial dust fluxes. However, we refrain from plotting the data because of its comparatively low temporal resolution for the targeted time interval.

7) Line 278: "Fe/Ca record lacks significant variability on the precession band". Statistical analyses are necessary (spectral analyses) to confirm this point.

Response: For a proper statistical analysis we have now performed a Blackman-Tukey spectral analyses for TOC, oxygen concentration, Fe/Al and Ti/Al ratios in comparison to the $\Delta$-insolation at 30°N. The reconstructed oxygen record of core SO-236-052 reveals strong power in the precession band (23 ka period). Significant but considerably weaker variability in the precession band is also detected in the TOC and Ti/Al records. The Fe/Al record lacks substantial precessional variability but is rather dominated by long-term glacial-interglacial changes. All of the above-mentioned results will be displayed and discussed in the revised manuscript.

8) Lines 336-337: "While the benthic foraminiferal fauna preliminary show changes on glacial-interglacial time scale, the TOC content and Ba/Ca ratio are characterized by additional variability in the precessional band." Again, statistical analyses are necessary (spectral analyses) to confirm this point.

Response: See response to referee comment #7.

9) Lines 342-347: "Elevated TOC and Ba/Ca ratios at site SO-236-052 during phases of reduced northern hemisphere summer insolation suggest a direct influence of the Indian winter monsoon on productivity and related organic matter fluxes of the Maldives Inner Sea during the past 200 ka, which is consistent with the present-day situation (de Vos et al. 2014). The close link between the winter monsoon intensity and surface

water productivity in the study area is confirmed by the difference between the $\delta$13C values of the epipelagic G. ruber (Gr) and the epibenthic C. mabahethi (Cm) (Figs. 3, 8)." Again, statistical analyses are necessary with a phase analyse. Also, what could be the role of the IEW and ENSO mentioned in the introduction part?

Response: With exception of the oxygen record, the TOC, Ti/Al, Fe/Al records reveal a comparatively weak coherence in the precession band (see also answer to referee comment #7). This result is likely related to the strong dominance of the 100 ka periodicity (as for example reflected in the dust supply) and superposition of shorter-wave variability. Hence, we did not include results from cross-spectral analyses. We have acknowledged the documented influence of IEW and ENSO variability on equatorial Indian surface ocean environments in the introduction chapter. On the other hand, the close relation of the present-day productivity in the Maldives Inner Sea surface waters (as reflected in seasonal satellite chlorophyll images) to the northern hemisphere winter season clearly demonstrates a relation to the NE monsoon. Specifically, our proxy records suggest enhanced dust fluxes and enhanced productivity during glacial boundary conditions underlining a general affiliation of Maldives paleoenvironments to the NE monsoon and high-latitude climate changes (dust availability, sea-level changes). To admit, we cannot exclude a potential additional influence of changes in IEW and ENSO but a proper statistical evaluation of phase relationships in the precessional band is unfortunately inhibited by the relatively weak precessional component and coherence in our proxy records of surface water productivity (such as TOC content, Br XRF counts). In the revised discussion chapter, we will address this issue in order to better acknowledge the possibility of IEW influence as observed in other studies.

10) Lines 372-376: "Long-term trends of similar magnitude have been recorded from sites bathed by the Antarctic Intermediate Water mass (AAIW) in the southwestern Pacific Ocean (Pahnke and Zahn, 2005; Elmore et al., 2015; Ronge et al., 2015). The general resemblance of the various epibenthic $\delta$13C records suggests a significant role of AAIW in ventilation of bathyal environments of the Maldives Inner Sea, which

is consistent with the modern oceanographic situation (You, 1998)." It would be good to add the data of the previous work mentioned on the Figure of the manuscript for a direct comparison.

Response: Thanks for raising this important point. For the revised version of our manuscript we will create an additional figure comparing our epibenthic stable carbon isotope record with published records of Pahnke and Zahn (2005), Elmore et al. (2015) and Ronge et al. (2015). The general resemblance of the $\delta13C$ trends from different regions confirms the super-regional AAIW influence.

11) Lines 385-386: "The reconstructed O2 record reveals precessional changes between oxic and low oxic conditions during northern hemisphere insolation maxima and minima, respectively". Statistical analyses are necessary (spectral analyses) to confirm this point.

Response: Spectral analysis of the oxygen record reveals significant variability in the precession band (see also answer to referee comment #7).

12) Lines 406-412: "Agulhas leakage". I do not understand this paragraph and the link with the Agulhas leakage. If the authors want to demonstrate a link between their record and the Indian monsoon they can compare directly with published monsoon records. Also, the forcing of the Agulhas leakage at terminations is driven by the subtropical front migration and is not directly link to the Indian monsoon (Peeters et al., 2004). For the IEW impact, a statistical analyze with the phase relationship (spectral analyses) will help the interpretation of the record.

Response: We admit, that the discussion related to the Agulhas leakage is not essential for the main conclusions of our paper. In order to avoid confusion, we will delete this paragraph. For the potential IEW impact see our response to referee comment #9.

References

Altabet, M. A., Higginson, M. J., and Murray, D. W.: The effect of millennial-scale

changes in Arabian Sea denitrification on atmospheric CO2, Nature, 415, 159–162, doi:10.1038/415159a, 2002.

Böning, P. and Bard, E.: Millennial/centennial-scale thermocline ventilation changes in the Indian Ocean as reflected by aragonite preservation and geochemical variations in Arabian Sea sediments, Geochimica et Cosmochimica Acta, 73, 6771–6788, doi:10.1016/j.gca.2009.08.028, 2009.

Boyd, P. W., Watson, A. J., Law, C. S., Abraham, E. R., Trull, T., Murdoch, R., Bakker, D. C. E., Bowie, A. R., Buesseler, K. O., Chang, H., Charette, M., Croot, P., Downing, K., Frew, R., Gall, M., Hadfield, M., Hall., J., Harvey, M., Jameson, G., LaRoche, J., Liddicoat, M., Ling, R., Maldonado, M. T., McKay, R. M., Nodder, S., Pickmere, S., Pridmore, R., Rintoul, S., Safi, K., Sutton, P., Strzepek, R., Tanneberger, K., Turner, S., Waite, A., and Zeldis, J.: A mesoscale phytoplankton bloom in the polar Southern Ocean simulated by iron fertilization, Nature, 407, 695–702, doi:10.1038/35037500, 2000.

Caley, T., Zaragosi, S., Bourget, J., Martinez, P., Malaizé, B., Eynaud, F., Rossignol, L., Garlan, T., and Ellouz-Zimmermann, N.: Southern Hemisphere imprint for Indo-Asian summer monsoons during the last glacial period as revealed by Arabian Sea productivity records, Biogeosciences, 10, 7347–7359, doi:10.5194/bg-10-7347-2013, 2013.

Clemens, S. C., Murray, D. W., and Prell, W. L.: Nonstationary Phase of the Plio-Pleistocene Asian monsoon, Science, 274, 943–948, 1996.

Deplazes, G., Lückge, A., Peterson, L. C., Timmermann, A., Hamann, Y., Hughen, K. A., Röhl, U., Laj, C., Cane, M. A., Sigman, D. M., and Haug, G. H.: Links between tropical rainfall, and North Atlantic climate during the last glacial period, Nature Geoscience, 6, 213–217, doi:10.1038/NGEO1712, 2013.

Dutta, K., Bhushan, R., and Somayajulu, B. L. K.: $\Delta$R Correction Values for the North-

Interactive
comment
ern Indian Ocean, Radiocarbon, 43, 483–488, doi:10.1017/S0033822200038376, 2001.

Elmore, A. C., McClymont, E. L., Elderfield, H., Kender, S., Cook, M. R., Leng, M. J., Greaves, M., and Misra, S.: Antarctic Intermediate Water properties since 400 ka recorded in infaunal (Uvigerina peregrina) and epifaunal (Planulina wuellerstorfi) benthic foraminifera, Earth and Planetary Science Letters, 428, 193–203, doi:10.1016/j.epsl.2015.07.013, 2015.

Gao, Y., Kaufman, Y. J., Tanré, D., Kolber, D., and Falkowski, P. G.: Seasonal Distribution of Aeolian Iron Fluxes to the Global Ocean, Geophysical Research Letters, 28, 29–32, 2001.

Itambi, A. C., von Dobeneck, T., Mulitza, S., Bickert, T., and Heslop, D.: Millennial-scale northwest African droughts related to Heinrich events and Dansgaard-Oeschger cycles: Evidence in marine sediments from offshore Senegal, Paleoceanography, 24, PA1205, 1–16, doi:10.1029/2007PA001570, 2009.

Jickells, T. D., An, Z. S., Andersen, K. K., Baker, A. R., Bergametti, G., Brooks, N., Cao, J. J., Boyd, P. W., Duce, R. A., Hunter, K. A., Kawahata, H., Kubilay, N., LaRoche, J., Liss, P. S., Mahowald, N., Prospero, J. M., Ridgwell, A. J., Tegen, I., and Torres, R.: Global Iron Connections Between Desert Dust, Ocean Biogeochemistry, and Climate, Science, 308, 67–71, doi:10.1126/science.1105959, 2005.

Lourens, L. J., Wehausen, R., and Brumsack, H.-J.: Geological constraints on tidal dissipation and dynamical ellipticity of the Earth over the past three million years, Nature, 409, 1029–1033, doi:10.1038/35059062, 2001.

Martin, J. H., Gordon, R. M., and Fitzwater, S. E.: The case for iron, Limnology and Oceanography, 36, 1793–1802, 1991.

Murray, J. W.: Ecology and Applications of Benthic Foraminifera, Cambridge University Press, 1–426, 2006.

Pahnke, K. and Zahn, R.: Southern hemisphere water mass conversion with North Atlantic climate variability, Science, 307, 1741–1746, doi:10.1126/science.1102163, 2005.

Pichevin, L., Bard, E., Martinez, P., and Billy, I.: Evidence of ventilation changes in the Arabian Sea during the late Quaternary: Implication for denitrification and nitrous oxide emission, Global Biogeochemical Cycles, 21, GB4008, 1–12, doi:10.1029/2006GB002852, 2007.

Ronge, T. A., Steph, S., Tiedemann, R., Prange, M., Merkel, U., Nürnberg, D., and Kuhn, G.: Pushing the boundaries: Glacial/interglacial variability of intermediate and deep waters in the southwest Pacific over the last 350,000 years, Paleoceanography, 30, 23–38, doi:10.1002/2014PA002727, 2015.

Southon, J., Kashgarian, M., Fontugne, M., Metivier, B., and Yim, W. W.-S.: Marine Reservoir Corrections for the Indian Ocean and Southeast Asia, Radiocarbon, 44, 167–180, doi:10.1017/S0033822200064778, 2002. Stuiver, M. and Reimer, P. J.: Extended 14C data base and revised CALIB 3.0 14C age calibration program, Radiocarbon, 35, 215–230, 1993.

Wehausen, R. and Brumsack, H.-J.: Chemical cycles in Pliocene sapropel-bearing and sapropel-barren eastern Mediterranean sediments, Palaeogeography, Palaeoclimatology, Palaeoecology, 158, 325–352, doi:10.1016/S0031-0182(00)00057-2, 2000.

Ziegler, M., Lourens, L. J., Tuenter, E., and Reichart, G.-J.: High Arabian Sea productivity conditions during MIS 13 – odd monsoon event or intensified overturning circulation at the end of the Mid-Pleistocene transition?, Climate of the past, 6, 63–76, 2010.

Zhang, X., Arimoto, R., An, Z., Chen, T., Zhang, G., Zhu, G., and Wang, X.: Atmospheric trace elements over source regions for Chinese dust: concentrations, sources and atmospheric depositions on the Loess Plateau, Atmospheric Environment, 27A, 2051–2067, doi:10.1016/0960-1686(93)90277-6, 1993.

---

## Author Comment (AC2) · 23 Jun 2017

Response to Referee#2

We acknowledge the detailed comments and suggestions by the reviewer, which helped to improve our manuscript considerably. In order to improve the statistical evaluation of our results we have performed Blackman-Tukey spectral analyses for selected proxy records. In addition, we have created new figures for BT power spectra and the comparison of own and published stable carbon isotope records. Below we respond to all comments raised by the reviewer.

With kind regards, Dorothea Bunzel

Overall, I think the analysis of the data and its presentation could be improved. Some

of the statements in the discussion (e.g. correlation of certain proxies with insolation or other proxy records, glacial-interglacial cycles) is often not supported by statistics or suitable figures (see comments below). I am also missing clear common thread and objective. This starts already in the abstract. It's starts of with a paragraph that basically says : "We measured a lot of stuff on a sediment core in the Maldives region. . .and then we interpreted the data..". I think it would be much more appealing if the manuscript would start with the context of the study, the main research question or problem or a hypothesis. Then they should list their approach (multi-proxy approach)

Response: Thank you for your suggestions. In the revised version we will include results from Blackman-Tukey spectral analysis in order to evaluate the variability in the precession band. For a proper statistical evaluation of long-term variations (i.e., in the excentricity band) our time series is too short. Our conclusions on the general glacial-interglacial variability are therefore still based on the graphical correlation among proxy records. We will rewrite the abstract putting our study in a general context and starting with the main objectives.

Further comments:

1) I think the paragraph from line 57 to line 66 could be improved. This paragraph contains a controversy in the interpretation of past OMZ variability in the Arabian Sea and its relation with summer monsoon variability. Is a strong OMZ linked to increased productivity (monsoon driven upwelling) or reduced ventilation (lower oxygen conc in southern sourced intermediate waters). The study by Bundel et al could inform this debate by providing a record of oxygen concentration from further South. There are some records (e.g. Ziegler et al., 2010, Climate of the Past) that show that a deep (most extended) OMZ occurs during glacial periods. While productivity maxima in the Arabian Sea, occur during interglacials. The new data by Bundel et al., could help to explain this observation by proving constraints of the Arabian Sea intermediate water ventilation from the South.

Response: According to the existing data and also supported by our new record, the OMZ in the Northern Indian Ocean is controlled by both changes in ventilation of inter­mediate (central OMZ; Das et al., in press 2017) and deep-water masses (deep OMZ; see Schmiedl and Leuschner, 2005; Ziegler et al., 2010), and by regional oxygen con­sumption responding to upwelling-driven high surface productivity (e.g., Reichart et al., 1998; Das et al., in press 2017). Our reconstruction corroborates this conclusion as it shows the general presence of an OMZ, which is also preconditioned by the biogeo­chemical processes in the Arabian Sea. The observation of generally reduced oxygen concentrations during glacial boundary conditions (particularly during MIS 2-4) reflects the ventilation signal of southern-derived intermediate water although with a probable regional signal of winter-monsoon-induced enhanced organic matter fluxes and oxygen consumption. The combination of the different factors (intermediate water circulation, summer and winter monsoon influence in different regions) will be emphasized and the discussion clarified in the revised manuscript.

2) Line 64: studies in stead of studied?

Response: All identified misspellings will be corrected in the revised manuscript.

3) line 77 -82: This section lists the main objectives of the study. Its strange that the objectives 1 and 2 mention suddenly, dust flux and sea-level, while the entire introduc­tion does not mention either of the two. I would focus on objective 3 and mention the subjects that deal with 1 and 2 in the discussion without putting to much emphasis on it.

Response: We agree that the role of sea level and dust fluxes is underrepresented in the introduction. Our proxy records highlight both changes in sea level and dust fluxes as relevant factors for sedimentation and marine ecosystem dynamics in the Maldives, Inner Sea. Therefore, we will enhance the introduction chapter including a new paragraph providing background information on the influence of these parameters in the wider study area.

4) line170: ". . .are based on. . ."

Response: It will be corrected.

5) line 173: 'was estimated to assess.."

Response: It will be corrected.

6) line 179-180: Why was a local reservoir age not applied?

Response: See response to referee#1 comment #3.

7) line 224: Given that the authors did XRF scanning, they should also have Bromine data. Bromine has been used successfully in several studies in the Indian Ocean as organic matter indicator (Caley et al., 2013, QSR, Ziegler et al, 2008, G3). The authors could do the same to get a high-resolution organic matter record and get a better idea of short term variability in TOC.

Response: Thank you for this suggestion. We have checked the bromine XRF counts in relation to the measured TOC values. Both data records reflect the same pronounced glacial-interglacial pattern with high values during glacial periods, but also reveal additional variability in the precession band over the studied time period. In the revised manuscript we will therefore include the Br XRF counts and will also evaluate it statistically (Blackman-Tukey spectral analysis). Both Br and Ba/Ca records show the same trend and both indicate marine productivity (Ziegler et al., 2008, 2009), but the Ba counts are comparatively low and therefore we will use the Br record instead of the Ba/Ca record in the revised manuscript.

8) line 258-260: What about the possibility that Fe/Ca and Si/Ca reflect changes in carbonate production / preservation? Maybe the dust input has been constant through time? See also related comments by the other reviewer. I fully agree with him.

Response: We agree since we cannot exclude potential changes in carbonate production. Therefore, we will include Ti/Al and Fe/Al instead of Fe/Ca as aeolian dust proxies

(see also answer to referee#1, comment #6). We will however still display and discuss the Si/Ca record since it reflects the availability of siliciclastic grains in the sediment for test construction of agglutinating foraminiferal species.

9) line 278: at the precessional band

Response: It will be corrected.

10) line 282: There are several studies that suggest that late Pleistocene quasi-100 kyr cycles are not driven by eccentricity, but instead are a response to skipped precession and/or obliquity cycles

Response: For a statistically more robust evaluation of the full orbital variability (including the long-wave components) in our data series we would need a time series, which is considerably longer than 200 ka. Nevertheless, graphical comparison of our data series reveals pronounced glacial-to-interglacial changes suggesting a link to eccentricity-driven environmental changes. Our conclusions are also in line with dust flux reconstructions from the Arabian Sea (e.g., Clemens et al., 1996) which show striking changes on the glacial-to-interglacial timescale (in the eccentricity band) suggesting a close link to environmental changes and associated dust availability on the northern borderlands. While the eccentricity component appears dominant in the dust proxies, variability in the precession band seems to be considerably lower (as indicated by spectral analyses). These results will be considered and discussed in the revised version.

11) line 342-345: I don't see a correlation of TOC or Ba with summer insolation. This should be demonstrated in a figure.

Response: In the revised version of figure 7 we will display Br XRF counts (see comment above) and TOC as indicators for productivity together with the difference of the summer and winter insolation at 30°N, which enables a graphical correlation of the mentioned proxy records and insolation. This comparison reveals coherent glacialto-interglacial changes in the TOC and Br record (also in the Ba/Ca record) with elevated values during glacial stages MIS 6 and MIS 2-4. For a statistical evaluation of the relation between insolation and the different proxy records we have performed a Blackman-Tukey spectral analyses and we will present the power spectra in a new figure.

12) section 4.2: This section seems not to be very important in the context of the whole manuscript. I would therefore again suggest to omit the sea-level topic from the list of main objectives.

Response: We are convinced that sea-level changes exert a strong impact on sedimentation processes and paleoenvironmental conditions of the Maldives Inner Sea. This is clearly reflected by the composition of the benthic foraminiferal fauna (e.g. assemblage 1, meroplanktonic taxa) and other parameters, such as the Sr/Ca ratio and grain size etc. We therefore do not want to omit this process from the main objectives. Instead, we will provide a bit more background on the relation between sea-level and Maldives paleoenvironments in the introduction chapter (see also comment above).

13) line 370-376: show the comparison with other datasets also in the figures otherwise the reader cannot judge your arguments

Response: We agree with this suggestion. We will create an additional figure for the epibenthic stable carbon isotope records, which will facilitate comparison of our data with published data (Pahnke and Zahn, 2005; Elmore et al., 2015; Ronge et al., 2015). See also our answer to referee#1, comment #10).

14) line 388-390: This sentence seems to contain a contradiction. Is the Maldives OMZ controlled by expansion of the Arabian Sea OMZ are controlled by the ventilation of southern sourced waters. (I would think it is the latter)

Response: The present OMZ of the northwestern Indian Ocean extends from the northern Arabian Sea into the tropical Indian Ocean (Reid, 2003) reflecting the reduced ventilation of intermediate water masses (due to its remote position) and the biogeochemical processes related to monsoon-induced organic matter fluxes and decomposition. We therefore assume that the OMZ variability in the Maldives Inner Sea is influenced by the overall strength and lateral expansion of the Arabian Sea OMZ, but it is additionally controlled by the ventilation of southern-derived oxygen-rich intermediate waters (AAIW) and by local monsoon-related organic matter fluxes and oxygen consumption. The general resemblance of our epibenthic stable carbon isotope record with comparable records from other areas indicates an ocean-wide link of intermediate water ventilation. On the other hand, the significant variability of our new oxygen reconstruction from the Maldives Inner Sea in the precession band and its resemblance with the reconstruction from the Arabian Sea suggests an additional influence of monsoon-driven biogeochemical processes. We will clarify the text accordingly.

15) line 396-401: I would argue the other way around. Low oxygen conc in intermediate waters in the Maldives area preconditioned the waters that ventilate the Arabian Sea. So a deep Arabian Sea OMZ has its root in the central Indian Ocean (and is thus not exclusively controlled by monsoon variability).

Response: see also comment above. The oxygen concentrations in the northwestern Indian Ocean display a gradient with very low values in the northern Arabian Sea and increasing values to the South. This gradient illustrates a clear relation to the monsoon-related biogeochemical processes in the Arabian Sea, but is also a reflection of the remote position of the Arabian Sea in terms of intermediate water ventilation. Nevertheless, a monsoon-induced strengthening of the OMZ in the Arabian Sea (as during MIS 3) will results in an increase of the north-south oxygen gradient in the entire northwestern Indian Ocean, which should then also be detected in the Maldives Inner Sea (although at a lower amplitude).

16) line 428: demonstrate cyclicality through spectral analysis (see also comment by other reviewer, fully agree)

Response: We have performed Blackman-Tukey spectral analyses and will present the results in the revised manuscript. See also answer to referee#1, comment #7.

17) Figure 6: Why is assemblage 2 abundant in the glacial MIS 6 and the Holocene? (Why is assemblage 1 abundant in 5, but absent in the Holocene)?

Response: This is a good question, but we do not yet have a simple explanation for it. Obviously, glacial conditions during MIS 6 were different from MIS 2-4 (Dansgaard et al., 1993); the latter was characterized by relatively lower sea-level and more intense glacial boundary conditions. Previous studies showed similar patterns, with certain benthic foraminiferal assemblages occurring both during glacial and interglacial periods, e.g. in the Red Sea (Badawi et al., 2005). At the Maldives Inner Sea glacial-to-interglacial changes in food fluxes were likely not extreme and therefore ecological thresholds for certain species and faunas may not have always been passed during glacial-interglacial transitions. A detailed inspection of assemblage 2 (C. mabahethi-fauna) actually displays faunal differences between their occurrences in MIS 1 and MIS 6 although C. mabahethi is the dominant taxon in both intervals.

References

Badawi, A., Schmiedl, G., and Hemleben, C.: Impact of late Quaternary environmental changes on deep-sea benthic foraminiferal faunas of the Red Sea, Marine Micropaleontology, 58, 13–30, doi:10.1016/j.marmicro.2005.08.002, 2005.

Clemens, S. C., Murray, D. W., and Prell, W. L.: Nonstationary Phase of the Plio-Pleistocene Asian monsoon, Science, 274, 943–948, 1996.

Dansgaard, W., Johnsen, S. J., Clausen, H. B., Dahl-Jensen, D., Gundestrup, N. S., Hammer, C. U., Hvidberg, C. S., Steffensen, J. P., Sveinbjörnsdottir, A. E., Jouzel, J., and Bond, G.: Evidence for general instability of past climate from a 250-kyr ice-core record, Nature, 364, 218–220, 1993.

Das, M., Singh, R. K., Gupta, A. K., and Bhaumik, A. K.: Holocene strengthening

of the oxygen minimum zone in the northwestern Arabian Sea linked to changes in intermediate water circulation or Indian monsoon intensity? Palaeogeography, Palaeoclimatology, Palaeoecology, doi:10.1016/j.palaeo.2016.10.035, in press 2017.

Elmore, A. C., McClymont, E. L., Elderfield, H., Kender, S., Cook, M. R., Leng, M. J., Greaves, M., and Misra, S.: Antarctic Intermediate Water properties since 400 ka recorded in infaunal (Uvigerina peregrina) and epifaunal (Planulina wuellerstorfi) benthic foraminifera, Earth and Planetary Science Letters, 428, 193–203, doi:10.1016/j.epsl.2015.07.013, 2015.

Pahnke, K. and Zahn, R.: Southern hemisphere water mass conversion with North Atlantic climate variability, Science, 307, 1741–1746, doi:10.1126/science.1102163, 2005.

Reichart, G. J., Lourens, L. J., and Zachariasse, W. J.: Temporal variability in the northern Arabian Sea Oxygen Minimum Zone (OMZ) during the last 225,000 years, Paleoceanography, 13, 607–621, doi:10.1029/98PA02203, 1998.

Reid, J. L.: On the total geostrophic circulation of the Indian Ocean: flow patterns, tracers, and transports, Progress in Oceanography, 56, 137–186, doi:10.1016/S0079-6611(02)00141-6, 2003.

Ronge, T. A., Steph, S., Tiedemann, R., Prange, M., Merkel, U., Nürnberg, D., and Kuhn, G.: Pushing the boundaries: Glacial/interglacial variability of intermediate and deep waters in the southwest Pacific over the last 350,000 years, Paleoceanography, 30, 23–38, doi:10.1002/2014PA002727, 2015.

Schmiedl, G. and Leuschner, D. C.: Oxygenation changes in the deep western Arabian Sea during the last 190,000 years: Productivity versus deepwater circulation, Paleoceanography, 20, PA2008, 1–14, doi:10.1029/2004PA001044, 2005.

Ziegler, M., Jilbert, T., de Lange, G., Lourens, L. J., and Reichart, G.-J.: Bromine counts from XRF scanning as an estimate of the marine organic carbon content of sediment

cores, Geochemistry, Geophysics, Geosystems, 9, 1–6, doi:10.1029/2007GC001932, 2008.

Ziegler, M., Lourens, L. J., Tuenter, E., and Reichart, G.-J.: Anomalously high Arabian Sea productivity conditions during MIS 13, Climate of the Past Discussions, 5, 1989–2018, 2009.

Ziegler, M., Lourens, L. J., Tuenter, E., and Reichart, G.-J.: High Arabian Sea productivity conditions during MIS 13 – odd monsoon event or intensified overturning circulation at the end of the Mid-Pleistocene transition?, Climate of the past, 6, 63–76, 2010.

---

## Author Response (AR1)

**Response to the editor decision and the reviews, concerning the discussion paper* "A multi-proxy analysis of late Quaternary Indian monsoon dynamics for the Maldives, Inner Sea"**

Dorothea Bunzel et al.

*Correspondence to:* dorothea.bunzel@uni-hamburg.de

**Dear Dr. Luc Beaufort,**

with this letter we submit a revised manuscript prepared for publication in Climate of the Past. The manuscript is entitled "A multi-proxy analysis of late Quaternary Indian monsoon dynamics for the Maldives, Inner Sea" and written by D. Bunzel, G. Schmiedl, S. Lindhorst, A. Mackensen, J. Reolid, S. Romahn, and C. Betzler. The manuscript has been prepared in accordance with the instruction for authors and none of the authors have any conflicts of interest (duplicate publication, financial, etc.).

We would like to thank you for acknowledging our replies to the reviews. The two anonymous reviewers are also thanked for bringing up their suggestion and substantial comments, which helped to improve our work considerably. We have changed the text accordingly and respond to all comments raised by the reviewers on the following pages. All relevant changes made in the manuscript (addressing lines, chapters or figures) are mentioned in our responses to the corresponding reviewer comments. Additionally, we have also made some minor revisions regarding misspellings, text phrasing, formatting of the text/figures, adding more values in the result chapters etc. We also suggest to provide the data records of sediment core SO236-052-4, which we used and discussed in our manuscript, to the PANGAEA Open Access library. We mentioned it in lines 211-212 at the end of the methods.

The structure on the next pages is as follows:

- Reviewer Comments (RC)
- *Authors Responses*
- Changes in the manuscript

*With kind regards (on behalf of all co-authors),*
*Dorothea Bunzel*

**Responses to Referee#1**

1) **RC#1:** I don't find the title of the current manuscript really suitable. The title suggest that the Maldives record is mainly driven by "Indian monsoon dynamics" whereas the authors conclude that the record provide a close linkage between the Indian monsoon oscillation, intermediate water circulation, productivity and sea-level changes on orbital time-scale. Therefore, a title such as "A multi-proxy analysis of late Quaternary equatorial Indian ocean for the Maldives, Inner Sea" could be less confusing.

   *Response: Thank you for suggesting a more suitable title. We have adjusted the title as follows: "A multi-proxy analysis of Late Quaternary ocean and climate variability for the Maldives, Inner Sea"*

2) **RC#1:** Lines 57 to 59. There is much more references of Arabian Sea works at the orbital and suborbital time scales (Clemens et al., 1996; Altabet et al., 2002; Clemens and Prell, 2003; Pichevin et al., 2007; Boning and Bard, 2009; Ziegler et al., 2010; Caley et al., 2011; Caley et al., 2013, Deplazes et al., 2013 are some examples).

   *Response: Thank you for recommending these works; we have included the mentioned references in our revised version (now lines 47-50).*

3) **RC#1:** Lines 179-180: "Local reservoir corrections were not applied". The authors should explain why they do not applied a correction. In general a correction of 400 years is applied in the tropics.

   *Response: We did not correct our radiocarbon ages for local reservoir effects, because of the contrast between the closest available numbers of marine reservoir age corrections, which are from the Arabian Sea, Northern Indian Ocean and Bay of Bengal, and between 821 and 864 km distance from our study site. We have changed the text as follows (now lines 203-206): "Due to the contrasting available reservoir age correction values (varying between 301 to 544 years; Dutta et al., 2001; Southon et al., 2002), the AMS $^{14}$C ages were corrected for the global reservoir age of 400 years in order to minimize potential errors and converted to calendar years using the radiocarbon calibration program CALIB (version 7.0.4; Stuiver and Reimer, 1993) and the calibration curve Marine13 (Reimer et al., 2013)."*

4) **RC#1:** Lines 202-209: Oxygen concentration should be shown on figure 3 and not only on Figure 8.

   *Response: We have included all full-resolution $\Delta\delta^{13}C$ data, which were used for the bottom-water oxygen reconstruction $(\Delta\delta^{13}C_{Cm-Ga})$ and water column mixing $(\Delta\delta^{13}C_{Gr-Cm})$ in Figure 3c.*

5) **RC#1:** Lines 217-221: The data of core M74/4-1143 are not shown on figure 4 making the comparison with core SO-236-052-4 impossible.

   *Response: We have included the sortable silt data of core M74/4-1143 in the Figure (Fig. 5a). Due to our additional created figures for the revised manuscript, the former Figure 4 became Figure 5 now.*

6) **RC#1:** Lines 228-233 and 258-259; Are the Fe/Ca and Si/Ca good proxies for Aeolian dust? The results could be compared to dust data from site ODP722 (Clemens et al., 1996). This is important to discuss the provenance of the dust and the interpretation of the Fe/Ca and Si/Ca that stays speculative in the discussion (lines 258-268). Also, previous study in the Arabian Sea used rather the changes in the Ti/Al ratio of the sediments as indicator for grain size and thus wind speed, since Titanium is concentrated in heavy minerals in the coarser size fraction (Reichart et al., 1997; Ziegler et al., 2010 CP).

   *Response: We consider the more commonly used proxies for terrigenous sediment delivery/aeolian dust fluxes and we have modified our manuscript as follows: We have replaced the Fe/Ca record by the Ti/Al and Fe/Al records as proxies for aeolian dust supply and enhanced aridity of the hinterland/source area (e.g. Zhang et al., 1993; Lourens et al.,*

*2001; Itambi et al., 2009) in order to account for a potential influence of changes in carbonate production and preservation on the Fe/Ca ratio (Wehausen and Brumsack, 2000). We have included the Fe/Al record because the aeolian Fe flux to the ocean likely has a direct impact on the seasonal surface ocean productivity (e.g. Martin et al., 1991; Boyd et al., 2000; Gao et al., 2001; Jickells et al., 2005), which also influences the deep-sea benthic ecosystems through seasonal phytodetritus pulses. Both Ti/Al and Fe/Al records show a similar glacial-interglacial pattern with relatively higher values during cold stages corroborating the foraminiferal results of enhanced surface ocean fertilisation of the Maldives Inner Sea during glacial periods (see lines 257-260, as well as Figs. 4-5, 8). We also show and discuss the Si/Ca ratio in addition, because we argue that the increase in agglutinated benthic foraminifera is a reflection of the availability of terrestrial particles, since most of these agglutinated species preferentially use siliciclastic grains for building up their test walls (e.g., Murray, 2006).*

*We have referred to the lithogenic flux record of site ODP722 (Clemens et al., 1996) since it supports our observation of generally enhanced glacial dust fluxes (for discussion see lines 298-309 and 321-323). However, we have refrained from plotting the data because of its comparatively low temporal resolution for the targeted time interval.*

7) **RC#1:** Line 278: "Fe/Ca record lacks significant variability on the precession band". Statistical analyses are necessary (spectral analyses) to confirm this point.

*Response: For a proper statistical analysis we have now performed a Blackman-Tukey spectral analyses for the oxygen concentrations of core SO236-052 and GeoB3004, as well as for TOC, bromine, Fe/Al and Ti/Al ratios of core SO236-052 in comparison to the Δ-insolation at 30° N (for methods see chapter 2.5: Spectral analyses). Both reconstructed oxygen records reveal strong power in the precession band (23 ka period); see lines 234-235. Significant but considerably weaker variability in the precession band is also detected in the TOC, Ti/Al records and the Br XRF counts. The Fe/Al record lacks substantial precessional variability but is rather dominated by long-term glacial-interglacial changes (lines 263-265). All results of the Blackman-Tukey spectral analysis are displayed in Fig. 4.*

8) **RC#1:** Lines 336-337: "While the benthic foraminiferal fauna preliminary show changes on glacial-interglacial time scale, the TOC content and Ba/Ca ratio are characterized by additional variability in the precessional band." Again, statistical analyses are necessary (spectral analyses) to confirm this point.

*Response: See response to referee comment #7.*

9) **RC#1:** Lines 342-347: "Elevated TOC and Ba/Ca ratios at site SO-236-052 during phases of reduced northern hemisphere summer insolation suggest a direct influence of the Indian winter monsoon on productivity and related organic matter fluxes of the Maldives Inner Sea during the past 200 ka, which is consistent with the present-day situation (de Vos et al. 2014). The close link between the winter monsoon intensity and surface water productivity in the study area is confirmed by the difference between the $\delta^{13}C$ values of the epipelagic G. ruber (Gr) and the epibenthic C. mabahethi (Cm) (Figs. 3, 8)." Again, statistical analyses are necessary with a phase analyse. Also, what could be the role of the IEW and ENSO mentioned in the introduction part?

*Response: With exception of the oxygen records, the TOC, Ti/Al, Fe/Al records reveal a comparatively weak coherence in the precession band (see also answer to referee comment #7). This result is likely related to the strong dominance of the 100 ka periodicity (as for example reflected in the dust supply) and superposition of shorter-wave variability. Hence, we did not include results from cross-spectral analyses.*

*We have acknowledged and discussed the documented influence of IEW and ENSO variability on equatorial Indian surface ocean environments in the introduction and the discussion chapters (lines 50-60 and 478-486). On the other hand, the close relation of the present-day productivity in the Maldives Inner Sea surface waters (as reflected in seasonal satellite chlorophyll a images) to the northern hemisphere winter season clearly demonstrates a relation to the*

*NE monsoon of this particular area. Specifically, our proxy records suggest enhanced dust fluxes and productivity during glacial boundary conditions underlining a general affiliation of Maldives Inner Sea paleoenvironments to the NE monsoon and high-latitude climate changes (dust availability, sea-level changes). To admit, we cannot exclude a potential additional influence of changes in IEW and ENSO but a proper statistical evaluation of phase relationships in the precessional band is unfortunately inhibited by the relatively weak precessional component and coherence in our proxy records of surface water productivity (such as TOC content, Br XRF counts; see Fig. 4). But we addressed this issue in our revised manuscript in order to better acknowledge the possibility of IEW influence as observed in other studies.*

10) **RC#1:** Lines 372-376: "Long-term trends of similar magnitude have been recorded from sites bathed by the Antarctic Intermediate Water mass (AAIW) in the southwestern Pacific Ocean (Pahnke and Zahn, 2005; Elmore et al., 2015; Ronge et al., 2015). The general resemblance of the various epibenthic $\delta^{13}$C records suggests a significant role of AAIW in ventilation of bathyal environments of the Maldives Inner Sea, which is consistent with the modern oceanographic situation (You, 1998)." It would be good to add the data of the previous work mentioned on the Figure of the manuscript for a direct comparison.

*Response: Thanks for raising this important point. For the revised version of our manuscript we have created an additional figure comparing our epibenthic stable carbon isotope record with the published records from the southwestern Pacific Ocean (Fig. 10): SO136/003 (Thiede et al., 1999), MD97-2120 (Pahnke and Zahn, 2005), DSDP593 (Elmore et al., 2015), MD06-2986 and MD06-2990 (Ronge et al., 2015). The general resemblance of the $\delta^{13}$C trends from different regions confirms the super-regional AAIW influence.*

11) **RC#1:** Lines 385-386: "The reconstructed $O_2$ record reveals precessional changes between oxic and low oxic conditions during northern hemisphere insolation maxima and minima, respectively". Statistical analyses are necessary (spectral analyses) to confirm this point.

*Response: Spectral analysis of the oxygen records reveals significant variability in the precession band (see also answer to referee comment #7 and Fig. 4a).*

12) **RC#1:** Lines 406-412: "Agulhas leakage". I do not understand this paragraph and the link with the Agulhas leakage. If the authors want to demonstrate a link between their record and the Indian monsoon they can compare directly with published monsoon records. Also, the forcing of the Agulhas leakage at terminations is driven by the subtropical front migration and is not directly link to the Indian monsoon (Peeters et al., 2004). For the IEW impact, a statistical analyze with the phase relationship (spectral analyses) will help the interpretation of the record.

*Response: We admit, that the discussion related to the Agulhas leakage is not essential for the main conclusions of our paper. In order to avoid confusion, we have deleted this paragraph. For the potential IEW impact see our response to referee comment #9.*

**Responses to Referee#2**

1) **RC#2:** Overall, I think the analysis of the data and its presentation could be improved. Some of the statements in the discussion (e.g. correlation of certain proxies with insolation or other proxy records, glacial-interglacial cycles) is often not supported by statistics or suitable figures (see comments below).

   I am also missing clear common thread and objective. This starts already in the abstract. It's starts of with a paragraph that basically says : "We measured a lot of stuff on a sediment core in the Maldives region. . .and then we interpreted the data..". I think it would be much more appealing if the manuscript would start with the context of the study, the main research question or problem or a hypothesis. Then they should list their approach (multi-proxy approach)

   *Response: Thank you for your suggestions. In the revised version we have included results from Blackman-Tukey spectral analysis in order to evaluate the variability in the precession band (see chapter 2.5: Spectral analyses). For a proper statistical evaluation of long-term variations (i.e., in the eccentricity band) our time series is too short. Our conclusions on the general glacial-interglacial variability are therefore still based on the graphical correlation among proxy records.*

   *We rewrote the first part of the abstract addressing the relevance and principal objectives of our study, and explaining why we selected the particular study area (see lines 9-15).*

2) **RC#2:** I think the paragraph from line 57 to line 66 could be improved. This paragraph contains a controversy in the interpretation of past OMZ variability in the Arabian Sea and its relation with summer monsoon variability. Is a strong OMZ linked to increased productivity (monsoon driven upwelling) or reduced ventilation (lower oxygen conc in southern sourced intermediate waters). The study by Bundel et al could inform this debate by providing a record of oxygen concentration from further South. There are some records (e.g. Ziegler et al., 2010, Climate of the Past) that show that a deep (most extended) OMZ occurs during glacial periods. While productivity maxima in the Arabian Sea, occur during interglacials. The new data by Bundel et al., could help to explain this observation by proving constraints of the Arabian Sea intermediate water ventilation from the South.

   *Response: According to the existing data and also supported by our new record, the OMZ in the northern Indian Ocean is controlled by both changes in ventilation of intermediate (central OMZ; Das et al., in press 2017) and deep-water masses (deep OMZ; see Schmiedl and Leuschner, 2005; Ziegler et al., 2010), and by regional oxygen consumption responding to upwelling-driven high surface productivity (e.g., Reichart et al., 1998; Das et al., in press 2017). Our reconstruction corroborates this conclusion as it shows the general presence of an OMZ, which is also preconditioned by the biogeochemical processes in the Arabian Sea. The observation of generally reduced oxygen concentrations during glacial boundary conditions (particularly during MIS 4-2) reflects the ventilation signal of southern-derived intermediate water although with a probable regional signal of winter-monsoon-induced enhanced organic matter fluxes and oxygen consumption. The combination of the different factors (intermediate water circulation, summer and winter monsoon influence in different regions) have been emphasized and the discussion clarified in the revised manuscript; see e.g. lines 88-93 and 449-459.*

3) **RC#2:** Line 64: studies in stead of studied?

   *Response: All identified misspellings have been corrected in the revised manuscript. The former sentences (lines 61-66, respectively) have been summarized (now lines 88-91).*

4) **RC#2:** line 77 -82: This section lists the main objectives of the study. Its strange that the objectives 1 and 2 mention suddenly, dust flux and sea-level, while the entire introduction does not mention either of the two. I would focus on objective 3 and mention the subjects that deal with 1 and 2 in the discussion without putting to much emphasis on it.

*Response: We agree that the role of sea-level and dust fluxes is underrepresented in the introduction. Our proxy records highlight both changes in sea-level and dust fluxes as relevant factors for sedimentation and marine ecosystem dynamics in the Maldives, Inner Sea. Therefore, we have enhanced the introduction chapter including a new paragraph providing background information on the influence of these parameters in the wider study area (lines 61-77).*

5) **RC#2:** line170: ". . .are based on. . ."

*Response: It has been corrected (line 190).*

6) **RC#2:** line 173: 'was estimated to assess..''

*Response: It has been corrected (line 192).*

7) **RC#2:** line 179-180: Why was a local reservoir age not applied?

*Response: See response to referee#1, comment #3, and the supplement we have made in the revised manuscript: lines 203-206.*

8) **RC#2:** line 224: Given that the authors did XRF scanning, they should also have Bromine data. Bromine has been used successfully in several studies in the Indian Ocean as organic matter indicator (Caley et al., 2013, QSR, Ziegler et al, 2008, G3). The authors could do the same to get a high-resolution organic matter record and get a better idea of short term variability in TOC.

*Response: Thank you for this suggestion. We have checked the bromine XRF counts in relation to the measured TOC values. Both data records reflect the same pronounced glacial-interglacial pattern with high values during glacial periods, but also reveal additional variability in the precession band over the studied time period (Figs. 4c-d, 8b-c). In the revised manuscript we have replaced the Ba/Ca record by the Br XRF counts and evaluated the latter statistically (Blackman-Tukey spectral analysis, chapter 2.5).*

9) **RC#2:** line 258-260: What about the possibility that Fe/Ca and Si/Ca reflect changes in carbonate production / preservation? Maybe the dust input has been constant through time? See also related comments by the other reviewer. I fully agree with him.

*Response: We agree since we cannot exclude potential changes in carbonate production. Therefore, we have included Ti/Al and Fe/Al instead of Fe/Ca as aeolian dust proxies (see also answer to referee#1, comment #6). However, we still display and discuss the Si/Ca record since it reflects the availability of siliciclastic grains in the sediment for test construction of agglutinating foraminiferal species (see lines 125-129 for all XRF proxy records used in the revised manuscript).*

10) **RC#2:** line 278: at the precessional band

*Response: It has been corrected (line 312).*

11) **RC#2:** line 282: There are several studies that suggest that late Pleistocene quasi-100 kyr cycles are not driven by eccentricity, but instead are a response to skipped precession and/or obliquity cycles

*Response: For a statistically more robust evaluation of the full orbital variability (including the long-wave components) in our data series we would need a time series, which is considerably longer than 200 ka. Nevertheless, graphical comparison of our data series reveals pronounced glacial-to-interglacial changes suggesting a link to eccentricity-driven environmental changes (e.g. Fig. 8). Our conclusions are also in line with dust flux reconstructions from the Arabian Sea (e.g., Clemens et al., 1996) which show striking changes on the glacial-to-interglacial timescale (in the*

*eccentricity band) suggesting a close link to environmental changes and associated dust availability on the northern borderlands. While the eccentricity component appears dominant in the dust proxies, variability in the precession band seems to be considerably lower (as indicated by spectral analyses; see Fig. 4). We considered and discussed these results in the revised version (e.g. lines 313-317 and 321-323).*

12) **RC#2:** line 342-345: I don't see a correlation of TOC or Ba with summer insolation. This should be demonstrated in a figure.

*Response: In the revised version of figure 8 (former Fig. 7) we have displayed the Br XRF counts (see comment above) and TOC as indicators for productivity together with the difference of the summer and winter insolation at 30° N, which enables a graphical correlation of the mentioned proxy records and northern hemisphere insolation. This comparison reveals coherent glacial-to-interglacial changes in the TOC and Br record (also in the Ba/Ca record) with elevated values during glacial stages MIS 6 and MIS 4-2. For a statistical evaluation of the relation between insolation and the TOC and Br records we have performed a Blackman-Tukey spectral analyses and have presented the power spectra in a new figure (Fig. 4).*

13) **RC#2:** section 4.2: This section seems not to be very important in the context of the whole manuscript. I would therefore again suggest to omit the sea-level topic from the list of main objectives.

*Response: We are convinced that sea-level changes exert a strong impact on sedimentation processes and paleoenvironmental conditions of the Maldives Inner Sea. This is clearly reflected by the composition of the benthic foraminiferal fauna (e.g. assemblage 1, meroplanktonic taxa) and other parameters, such as the Sr/Ca ratio and grain size etc. We therefore did not omit this process from the main objectives. Instead, we provide a bit more background on the relation between sea-level and the sedimentary system/paleoenvironment of the Maldives in the introduction chapter (see also comment above, and lines 73-79).*

14) **RC#2:** line 370-376: show the comparison with other datasets also in the figures otherwise the reader cannot judge your arguments

*Response: We agree with this suggestion. We have created an additional figure for the epibenthic stable carbon isotope records (Fig. 10), which enables the comparison of our data with published data (Thiede et al., 1999; Pahnke and Zahn, 2005; Elmore et al., 2015; Ronge et al., 2015). See also our answer to referee#1, comment #10).*

15) **RC#2:** line 388-390: This sentence seems to contain a contradiction. Is the Maldives OMZ controlled by expansion of the Arabian Sea OMZ are controlled by the ventilation of southern sourced waters. (I would think it is the latter)

*Response: The present OMZ of the northwestern Indian Ocean extends from the northern Arabian Sea into the tropical Indian Ocean (Reid, 2003) reflecting the reduced ventilation of intermediate water masses (due to its remote position) and the biogeochemical processes related to monsoon-induced organic matter fluxes and decomposition. We therefore assume that the OMZ variability in the Maldives Inner Sea is influenced by the overall strength and lateral expansion of the Arabian Sea OMZ, but it is additionally controlled by the ventilation of southern-derived oxygen-rich intermediate waters (AAIW) and by local monsoon-related organic matter fluxes and oxygen consumption. The general resemblance of our epibenthic stable carbon isotope record with comparable records from other areas indicates an ocean-wide link of intermediate water ventilation. On the other hand, the significant variability of our new oxygen reconstruction from the Maldives Inner Sea in the precession band and its resemblance with the reconstruction from the Arabian Sea suggests an additional influence of monsoon-driven biogeochemical processes. We have clarified the text accordingly (e.g. lines 444-459).*

16) **RC#2:** line 396-401: I would argue the other way around. Low oxygen conc in intermediate waters in the Maldives area preconditioned the waters that ventilate the Arabian Sea. So a deep Arabian Sea OMZ has its root in the central Indian Ocean (and is thus not exclusively controlled by monsoon variability).

*Response: see also comment above. The oxygen concentrations in the northwestern Indian Ocean display a gradient with very low values in the northern Arabian Sea and increasing values to the South. This gradient illustrates a clear relation to the monsoon-related biogeochemical processes in the Arabian Sea, but is also a reflection of the remote position of the Arabian Sea in terms of intermediate water ventilation. Nevertheless, a monsoon-induced strengthening of the OMZ in the Arabian Sea (as during MIS 3) will results in an increase of the north-south oxygen gradient in the entire northwestern Indian Ocean, which should then also be detected in the Maldives Inner Sea (although at a lower amplitude). See lines 471-477.*

17) **RC#2:** line 428: demonstrate cyclicality through spectral analysis (see also comment by other reviewer, fully agree)

*Response: We have performed Blackman-Tukey spectral analyses and have included the results in the revised manuscript. See also answer to referee#1, comment #7, and chapter 2.5.*

18) **RC#2:** Figure 6: Why is assemblage 2 abundant in the glacial MIS 6 and the Holocene? (Why is assemblage 1 abundant in 5, but absent in the Holocene)?

*Response: This is a good question, but we do not yet have a simple explanation for it. Obviously, glacial conditions during MIS 6 were different from MIS 4-2 (Dansgaard et al., 1993); the latter was characterized by relatively lower sea-level and more intense glacial boundary conditions.*

[revised manuscript text omitted]

Figure 8

[Figure]

1000